# Network Diffusions via Neural Mean-Field Dynamics

**Shushan He**
Mathematics & Statistics
Georgia State University
Atlanta, Georgia, USA
she4@gsu.edu

**Hongyuan Zha**
School of Data Science
Shenzhen Research Institute of
Big Data, CUHK, Shenzhen, China
zhahy@cuhk.edu.cn

**Xiaojing Ye**
Mathematics & Statistics
Georgia State University
Atlanta, Georgia, USA
xye@gsu.edu

## Abstract

We propose a novel learning framework based on neural mean-field dynamics for *simultaneous* inference and estimation problems of diffusions on networks. Our new framework is derived from the Mori-Zwanzig formalism to obtain an exact evolution of the node infection probabilities, which renders a delay differential equation with memory integral approximated by learnable time convolution operators, resulting in a highly structured and interpretable RNN. Directly using cascade data, our framework can *jointly* learn the structure of the diffusion network and the evolution of infection probabilities, which are cornerstone to important downstream applications such as influence maximization. Connections between parameter learning and optimal control are also established. Empirical study shows that our approach is versatile and robust to variations of the underlying diffusion network models, and significantly outperforms existing approaches in accuracy and efficiency on both synthetic and real-world data.

## 1 Introduction

Continuous-time information diffusion on heterogeneous networks is a prevalent phenomenon [4, 39, 43]. News spreading on social media [13, 15, 49], viral marketing [23, 25, 52], computer malware propagation, and epidemics of contagious diseases [3, 36, 43, 47] are all examples of diffusion on networks, among many others. For instance, a piece of information (such as a tweet) can be retweeted by users (nodes) with followee-follower relationships (edge) on the Twitter network. We call a user *infected* if she retweets, and her followers see her retweet and can also become infected if they retweet in turn, and so on. Such information diffusion mimics the epidemic spread where an infectious virus can spread to individuals (human, animal, or plant) and then to many others upon their close contact.

In this paper, we are mainly concerned with the estimation of individual node infection probabilities as well as inference of the underlying diffusion network structures directly using cascade data of historical diffusion events on the network. For infection probability estimation, our goal is to compute the evolution of the probability of each node being infected during a diffusion initiated from a set of source nodes. For network structure inference, we aim at learning the edges as well as the strength of interactions (through the edges) between the nodes on the diffusion network. Not surprisingly, both problems are very challenging due to the extremely large scale of modern networks, the heterogeneous inter-dependencies among the nodes, and the randomness exhibited in cascade data. Most existing works focus on one problem only, e.g., either to solely infer the network structure from cascade data, or to estimate influence without providing insights into the underlying network structure.

We propose a novel learning framework, called neural mean-field (NMF) dynamics, to *simultaneously* tackle both of the estimation and inference problems mentioned above. Specifically: (i) We develop a neural mean-field dynamics framework to model the evolution of diffusion on a network. Our new framework is derived from the Mori-Zwanzig formalism to obtain an *exact* time evolution of the node infection probability with dimension linear in the network size; (ii) We show that the memory term of

the Mori-Zwanzig equation can be approximated by a trainable convolution network, which renders the dynamical system into a delay differential equation. We also show that the time discretization of such system reduces to a recurrent neural network. The approximate system is highly *interpretable*, and in particular, the training accepts sample cascades as input, and returns both individual probability estimates (and hence the influence function) as well as structure information of the diffusion network as outputs; (iii) We show that the parameters learning in NMF can be reduced to an optimal control problem with the parameters as time invariant control inputs, maximizing the *total Hamiltonian* of the system; and (iv) Our empirical analysis shows that our approach is robust to the variation of the unknown underlying diffusion models, and it also significantly outperforms existing approaches for both synthetic and real-world diffusion networks.

The remainder of this paper is organized as follows. In Section 2, we introduce the diffusion network models and related background information, including the influence predication and structure inference problems. In Section 3, we develop the proposed framework of neural mean-field dynamics for inference and prediction on diffusion networks, as well as an optimal control formulation for parameter learning. We demonstrate the performance of the proposed method on influence estimation and maximization on a variety of synthetic and real-world networks in Section 4. A discussion of the related work is given in Section 5. Section 6 concludes the paper.

## 2 Preliminaries on Diffusion Networks

Throughout this paper, we use boldfaced lower (upper) letter to denote vector (matrix) or vector-valued (matrix-valued) function, and $(\cdot)_k$ (or $(\cdot)_{ij}$) for its $k$th component (or $(i,j)$-th entry). All vectors are column vectors unless otherwise noted. We follow the Matlab syntax and use $[\boldsymbol{x}; \boldsymbol{y}]$ to denote the vector that stacks $\boldsymbol{x}$ and $\boldsymbol{y}$ vertically. We denote inner product by $\boldsymbol{x} \cdot \boldsymbol{y}$ and component-wise multiplication by $\boldsymbol{x} \odot \boldsymbol{y}$. Time is denoted by $t$ in either continuous ($t \in [0,T]$) or discrete case ($t = 0, 1, \ldots, T$) for some time horizon $T \in \mathbb{R}_+$ ($\mathbb{N}$ in discrete case). Derivative $'$ is with respect to $t$, and gradient $\nabla_{\boldsymbol{x}}$ is with respect to $\boldsymbol{x}$. Probability is denoted by $\Pr(\cdot)$, and expectation with respect to $X$ (or $p_X$) is denoted by $\mathbb{E}_X[\,\cdot\,]$.

**Diffusion network models**  Consider a diffusion network model, which consists of a network (directed graph) $\mathcal{G} = (\mathcal{V}, \mathcal{E})$ with node set $\mathcal{V} = [n] := \{1, \ldots, n\}$ and edge set $\mathcal{E} \subset \mathcal{V} \times \mathcal{V}$, and a *diffusion model* that describes the distribution $p(t; \alpha_{ij})$ of the time $t$ node $i$ takes to infect a healthy neighbor $j \in \{j' : (i, j') \in \mathcal{E}\}$ for every $(i,j) \in \mathcal{E}$. Then, given a source (seed) set $\mathcal{S}$ of nodes that are infected at time 0, they will infect their healthy neighbors with infection time following $p$, and the infected neighbors will then infect their healthy neighbors, and so on, such that the infection initiated from $\mathcal{S}$ at time 0 propagates to other nodes of the network.

Typical diffusion network models are assumed to be *progressive* where infected node cannot recover and the infections on different edges are independent. For example, the standard diffusion model with exponential distribution $p(t; \alpha) = \alpha e^{-\alpha t}$ is mostly widely used; other distributions can also be considered, as is done in this paper. For simplicity, we focus on uni-parameter distributions or distributions with multiple parameters but only one can vary across different edges with the consequence that the parameter $\alpha_{ij} \geq 0$ indicates the *strength* of impact node $i$ has on node $j$.

**Cascade data**  Observation data $\mathcal{D}$ of a diffusion network are often in the form of *sample cascades* $\mathcal{D} := \{\mathcal{C}_k = (\mathcal{S}_k, \boldsymbol{\tau}_k) \in 2^{\mathcal{V}} \times \mathbb{R}_+^n : k \in [K]\}$, where the $k$th cascade $\mathcal{C}_k$ records its source set $\mathcal{S}_k \subset \mathcal{V}$ and the time $(\boldsymbol{\tau}_k)_i \geq 0$ which indicates when node $i$ was infected (if $i$ was not infected during $\mathcal{C}_k$ then $(\boldsymbol{\tau}_k)_i = \infty$). We also equate $\mathcal{C}_k$ with $\{\hat{\boldsymbol{x}}^{(k)}(t) \in \{0,1\}^n : i \in [n], t \geq 0\}$ such that $(\hat{\boldsymbol{x}}^{(k)}(t))_i = 1$ if the node $i$ is in the infected status at time $t$ and 0 otherwise. For example, $\hat{\boldsymbol{x}}^{(k)}(0) = \boldsymbol{\chi}_{\mathcal{S}_k}$ where $(\boldsymbol{\chi}_{\mathcal{S}_k})_i = 1$ if $i \in \mathcal{S}_k$ and 0 otherwise. Such cascade data are collected from historical events for training purposes.

**Influence prediction and inference of diffusion network**  Given the network $\mathcal{G} = (\mathcal{V}, \mathcal{E})$, as well as the diffusion model and $\boldsymbol{A}$, where $(\boldsymbol{A})_{ji} = \alpha_{ij}$ is the parameter of $p(t; \alpha_{ij})$ for edge $(i,j)$, the *inference prediction* (or *influence estimation*) is to compute

$$\boldsymbol{x}(t; \boldsymbol{\chi}_{\mathcal{S}}) = [x_1(t; \boldsymbol{\chi}_{\mathcal{S}}), \ldots, x_n(t; \boldsymbol{\chi}_{\mathcal{S}})]^\top \in [0,1]^n \tag{1}$$

for all time $t \geq 0$ and any source set $\mathcal{S} \subset \mathcal{V}$. In (1), $x_i(t; \chi_\mathcal{S})$ is the probability of node $i$ being infected at time $t$ given a source set $\mathcal{S}$ (not necessarily observed as a source set in $\mathcal{D}$). Note that we use $\chi_\mathcal{S}$ and $\mathcal{S}$ interchangeably hereafter. The probability $\boldsymbol{x}(t; \chi_\mathcal{S})$ can also be used to compute the *influence function* $\sigma(t; \mathcal{S}) := \mathbf{1}_n^\top \boldsymbol{x}(t; \chi_\mathcal{S})$, the expected number of infected nodes at time $t$. Note that an analytic solution of (1) is intractable due to the exponentially large state space of the complete dynamical system of the diffusion problem [19, 48].

On the other hand, network *inference* refers to learning the network connectivity $\mathcal{E}$ and $\boldsymbol{A}$ given cascade data $\mathcal{D}$. The matrix $\boldsymbol{A}$ is the distribution parameters if the diffusion model $p$ is given, or it simply qualitatively measures the strength of impact node $i$ on $j$ if no specific $p$ is known.

Influence prediction may also require network inference when only cascade data $\mathcal{D}$ are available, resulting in a *two-stage* approach: a network inference is performed first to learn the network structure $\mathcal{E}$ and the diffusion model parameters $\boldsymbol{A}$, and then an influence estimation is used to compute the influence for the source set $\mathcal{S}$. However, approximation errors and biases in the two stages will certainly accumulate. Alternatively, one can use a *one-stage* approach to directly estimate $\boldsymbol{x}(t; \chi_\mathcal{S})$ of any $\mathcal{S}$ from the cascade data $\mathcal{D}$, which is more versatile and less prone to diffusion model misspecification. Our method is a such kind of one-stage method. Additionally, it allows knowledge of $\mathcal{E}$ and/or $\boldsymbol{A}$, if available, to be integrated for further performance improvement.

**Influence maximization**  Given cascade data $\mathcal{D}$, *influence maximization* is to find the source set $\mathcal{S}$ that generates the maximal influence $\sigma(t; \mathcal{S})$ at $t$ among all subsets of size $n_0$, where $t > 0$ and $1 \leq n_0 < n$ are prescribed. Namely, influence maximization can be formulated as

$$\max_{\mathcal{S}} \ \sigma(t; \mathcal{S}), \quad \text{s.t.} \quad \mathcal{S} \subset \mathcal{V}, \quad |\mathcal{S}| \leq n_0. \tag{2}$$

There are two main ingredients of an influence maximization method for solving (2): an influence prediction subroutine that evaluates the influence $\sigma(t; \mathcal{S})$ for any given source set $\mathcal{S}$, and an (approximate) combinatorial optimization solver to find the optimal set $\mathcal{S}$ of (2) that repeatedly calls the subroutine. The combinatorial optimization problem is NP-hard and is often approximately solved by greedy algorithms with guaranteed sub-optimality when $\sigma(t; \mathcal{S})$ is submodular in $\mathcal{S}$. In our experiment, we show that a standard greedy approach equipped with our proposed influence estimation method outperforms other state-of-the-art influence maximization algorithms.

## 3  Neural Mean-Field Dynamics

**Modelling diffusion by stochastic jump processes**  We begin with the jump process formulation of network diffusion. Given a source set $\chi_\mathcal{S}$, let $X_i(t; \chi_\mathcal{S})$ denote the infection status of the node $i$ at time $t$. Namely, $X_i(t) = 1$ if node $i$ is infected by time $t$, and 0 otherwise. Then $\{X_i(t) : i \in [n]\}$ are a set of $n$ coupled jump processes, such that $X_i(t; \chi_\mathcal{S})$ jumps from 0 to 1 when the node $i$ is infected by any of its infected neighbors at $t$. Let $\lambda_i^*(t)$ be the conditional intensity of $X_i(t; \chi_\mathcal{S})$ given the history $\mathcal{H}(t) = \{X_i(s; \chi_\mathcal{S}) : s \leq t, \ i \in [n]\}$, i.e.,

$$\lambda_i^*(t) := \lim_{\tau \to 0^+} \frac{\mathbb{E}[X_i(t + \tau; \chi_\mathcal{S}) - X_i(t; \chi_\mathcal{S}) | \mathcal{H}(t)]}{\tau}. \tag{3}$$

Note that the numerator of (3) is also the conditional probability $\Pr(X_i(t + \tau) = 1, X_i(t) = 0 | \mathcal{H}(t))$ for any $\tau > 0$. In influence prediction, our goal is to compute the probability $\boldsymbol{x}(t; \chi_\mathcal{S}) = [x_i(t; \chi_\mathcal{S})]$ in (1), which is the expectation of $X_i(t; \chi_\mathcal{S})$ conditioning on $\mathcal{H}(t)$:

$$x_i(t; \chi_\mathcal{S}) = \mathbb{E}_{\mathcal{H}(t)}[X_i(t; \chi_\mathcal{S}) | \mathcal{H}(t)]. \tag{4}$$

To this end, we adopt the following notations (for notation simplicity we temporarily drop $\chi_\mathcal{S}$ in this subsection as the source set $\mathcal{S}$ is arbitrary but fixed):

$$x_I(t) = \mathbb{E}_{\mathcal{H}(t)} \left[ \prod_{i \in I} X_i(t; \chi_\mathcal{S}) \big| \mathcal{H}(t) \right], \quad y_I(t) = \prod_{i \in I} x_i(t), \quad e_I(t) = x_I(t) - y_I(t) \tag{5}$$

for any $I \subset [n]$ and $|I| \geq 2$. Then we can derive the evolution of $\boldsymbol{z} := [\boldsymbol{x}; \boldsymbol{e}]$. Here $\boldsymbol{x}(t) \in [0, 1]^n$ is the *resolved* variable whose value is of interests and samples can be directly observed from the cascade data $\mathcal{D}$, and $\boldsymbol{e}(t) = [\cdots, e_I(t), \cdots] \in \mathbb{R}^{N-n}$ where $N = 2^n - 1$ is the *unresolved* variable that captures all the second and higher order moments. The complete evolution equation of $\boldsymbol{z}$ is given in the following theorem, where the proof is provided in Appendix B.1.

**Theorem 1.** *The evolution of $\boldsymbol{z}(t) = [\boldsymbol{x}(t); \boldsymbol{e}(t)]$ follows the nonlinear differential equation:*

$$\boldsymbol{z}' = \bar{\boldsymbol{f}}(\boldsymbol{z}), \quad \text{where} \quad \bar{\boldsymbol{f}}(\boldsymbol{z}) = \bar{\boldsymbol{f}}(\boldsymbol{x}, \boldsymbol{e}) = \left[\boldsymbol{f}(\boldsymbol{x}; \boldsymbol{A}) - (\boldsymbol{A} \odot \boldsymbol{E})\mathbf{1}; \cdots, f_I(\boldsymbol{x}, \boldsymbol{e}); \cdots\right], \quad (6)$$

*with initial value $\boldsymbol{z}_0 = [\boldsymbol{\chi}_{\mathcal{S}}; \mathbf{0}] \in \mathbb{R}^N$, $\boldsymbol{E} = [e_{ij}] \in \mathbb{R}^{n \times n}$, and*

$$\boldsymbol{f}(\boldsymbol{x}; \boldsymbol{A}) = \boldsymbol{A}\boldsymbol{x} - \mathrm{diag}(\boldsymbol{x})\boldsymbol{A}\boldsymbol{x}, \tag{7}$$

$$f_I(\boldsymbol{x}, \boldsymbol{e}) = \sum_{i \in I} \sum_{j \notin I} \alpha_{ji}(y_I - y_{I \cup \{j\}} + e_I - e_{I \cup \{j\}}) - \sum_{i \in I} y_{I \setminus \{i\}} \sum_{j \neq i} \alpha_{ji}(x_j - y_{ij} - e_{ij}). \tag{8}$$

The evolution (6) holds true exactly for the standard diffusion model with exponential distribution, but also approximates well for other distributions $p$, as shown in the empirical study below. In either case, the dimension $N$ of $\boldsymbol{z}$ grows exponentially fast in network size $n$ and hence renders the computation infeasible in practice. To overcome this issue, we employ the Mori-Zwanzig formalism [7] to derive a reduced-order model of $\boldsymbol{x}$ with dimensionality $n$ only.

**Mori-Zwanzig memory closure**   We employ the Mori-Zwanzig (MZ) formalism [7] that allows to introduce a generalized Langevin equation (GLE) of the $\boldsymbol{x}$ part of the dynamics (6). The GLE of $\boldsymbol{x}$ is derived from the original equation (6) describing the evolution of $\boldsymbol{z} = [\boldsymbol{x}; \boldsymbol{e}]$, while maintaining the effect of the unresolved part $\boldsymbol{e}$. This is particularly useful in our case, as we only need $\boldsymbol{x}$ for infection probability estimation and influence prediction.

Define the Liouville operator $\mathcal{L}$ such that $\mathcal{L}[g](\boldsymbol{z}) := \bar{\boldsymbol{f}}(\boldsymbol{z}) \cdot \nabla_{\boldsymbol{z}} g(\boldsymbol{z})$ for any real-valued function $g$ of $\boldsymbol{z}$. Let $e^{t\mathcal{L}}$ be the Koopman operator associated with $\mathcal{L}$ such that $e^{t\mathcal{L}} g(\boldsymbol{z}(0)) = g(\boldsymbol{z}(s))$ where $\boldsymbol{z}(t)$ solves (6). Then $\mathcal{L}$ is known to satisfy the semi-group property for all $g$, i.e., $e^{t\mathcal{L}} g(\boldsymbol{z}) = g(e^{t\mathcal{L}}\boldsymbol{z})$. Now consider the projection operator $\mathcal{P}$ as the truncation such that $(\mathcal{P}g)(\boldsymbol{z}) = (\mathcal{P}g)([\boldsymbol{x}; \boldsymbol{e}]) = g([\boldsymbol{x}; \mathbf{0}])$ for any $\boldsymbol{z} = [\boldsymbol{x}; \boldsymbol{e}]$, and its orthogonal complement as $\mathcal{Q} = I - \mathcal{P}$ where $I$ is the identity operator. The following theorem describes the *exact* evolution of $\boldsymbol{x}(t)$, and the proof is given in Appendix B.2.

**Theorem 2.** *The evolution of $\boldsymbol{x}$ specified in (6) can also be described by the following GLE:*

$$\boldsymbol{x}' = \boldsymbol{f}(\boldsymbol{x}; \boldsymbol{A}) + \int_0^t \boldsymbol{k}(t - s, \boldsymbol{x}(s)) \, \mathrm{d}s, \tag{9}$$

*where $\boldsymbol{f}$ is given in (7), and $\boldsymbol{k}(t, \boldsymbol{x}) := \mathcal{P}\mathcal{L}e^{t\mathcal{Q}\mathcal{L}}\mathcal{Q}\mathcal{L}\boldsymbol{x}$.*

Note that, (9) is *not* an approximation—it is an *exact* representation of the $\boldsymbol{x}$ part of the original problem (6). The equation (9) can be interpreted as a *mean-field* equation, where the two terms on the right hand side are called the *streaming term* (corresponding to the mean-field dynamics) and *memory term*, respectively. The streaming term provides the *main drift* of the evolution, and the memory term in the convolution form is for vital *adjustment*. This inspires us to approximate the memory term as a time convolution on $\boldsymbol{x}$, which naturally yields a delay differential equation and further reduces to a structured recurrent neural network (RNN) after discretization, as shown in the next subsection.

**Delay differential equation and RNN**   To compute the evolution (9) of $\boldsymbol{x}$, we consider an approximation of the Mori-Zwanzig memory term by a neural net $\boldsymbol{\varepsilon}$ with time convolution of $\boldsymbol{x}$ as follows,

$$\int_0^t \boldsymbol{k}(t - s, \boldsymbol{x}(s)) \, \mathrm{d}s \approx \boldsymbol{\varepsilon}(\boldsymbol{x}(t), \boldsymbol{h}(t); \boldsymbol{\eta}) \quad \text{where} \quad \boldsymbol{h}(t) = \int_0^t \boldsymbol{K}(t - s; \boldsymbol{w})\boldsymbol{x}(s) \, \mathrm{d}s. \tag{10}$$

In (10), $\boldsymbol{K}(\cdot; \boldsymbol{w})$ is a convolutional operator with parameter $\boldsymbol{w}$, and $\boldsymbol{\varepsilon}(\boldsymbol{x}, \boldsymbol{h}; \boldsymbol{\eta})$ is a deep neural net with $(\boldsymbol{x}, \boldsymbol{h})$ as input and $\boldsymbol{\eta}$ as parameter. Both $\boldsymbol{w}$ and $\boldsymbol{\eta}$ are to be trained by the cascade data $\mathcal{D}$. Hence, (9) reduces to a *delay differential equation* which involves a time integral $\boldsymbol{h}(t)$ of past $\boldsymbol{x}$:

$$\boldsymbol{x}' = \tilde{\boldsymbol{f}}(\boldsymbol{x}, \boldsymbol{h}; \boldsymbol{\theta}) := \boldsymbol{f}(\boldsymbol{x}; \boldsymbol{A}) + \boldsymbol{\varepsilon}(\boldsymbol{x}, \boldsymbol{h}; \boldsymbol{\eta}). \tag{11}$$

The initial condition of (11) with source set $\mathcal{S}$ is given by

$$\boldsymbol{x}(0) = \boldsymbol{\chi}_{\mathcal{S}}, \quad \boldsymbol{h}(0) = \mathbf{0}, \quad \text{and} \quad \boldsymbol{x}(t) = \boldsymbol{h}(t) = \mathbf{0}, \quad \forall t < 0. \tag{12}$$

We call the system (11) with initial (12) the *neural mean-field* (NMF) dynamics.

The delay differential equation (11) is equivalent to a coupled system of $(\boldsymbol{x}, \boldsymbol{h})$. In addition, we show that the discretization of this system reduces to a structured recurrent neural network if $\boldsymbol{K}(t; \boldsymbol{w})$ is a (linear combination of) matrix convolutions in the following theorem.

**Theorem 3.** *The delay differential equation* (11) *is equivalent to the following coupled system:*

$$\boldsymbol{x}' = \tilde{\boldsymbol{f}}(\boldsymbol{x}, \boldsymbol{h}; \boldsymbol{A}, \boldsymbol{\eta}) = \boldsymbol{f}(\boldsymbol{x}; \boldsymbol{A}) + \boldsymbol{\varepsilon}(\boldsymbol{x}, \boldsymbol{h}; \boldsymbol{\eta}) \tag{13a}$$

$$\boldsymbol{h}' = \int_0^t \boldsymbol{K}(t - s; \boldsymbol{w}) \tilde{\boldsymbol{f}}(\boldsymbol{x}(s), \boldsymbol{h}(s); \boldsymbol{A}, \boldsymbol{\eta}) \, \mathrm{d}s \tag{13b}$$

*with initial condition* (12). *In particular, if* $\boldsymbol{K}(t; \boldsymbol{w}) = \sum_{l=1}^{L} \boldsymbol{B}_l e^{-\boldsymbol{C}_l t}$ *for some* $L \in \mathbb{N}$ *with* $\boldsymbol{w} = \{(\boldsymbol{B}_l, \boldsymbol{C}_l)_l : \boldsymbol{B}_l \boldsymbol{C}_l = \boldsymbol{C}_l \boldsymbol{B}_l, \forall l \in [L]\}$, *then* (13) *can be solved by a non-delay system of* $(\boldsymbol{x}, \boldsymbol{h})$ *with* (13a) *and* $\boldsymbol{h}' = \sum_{l=1}^{L} (\boldsymbol{B}_l \boldsymbol{x} - \boldsymbol{C}_l \boldsymbol{h})$. *The discretization of such system (with step size normalized to 1) reduces to an RNN with hidden layers* $(\boldsymbol{x}_t, \boldsymbol{h}_t)$ *for* $t = 0, 1, \ldots, T-1$:

$$\boldsymbol{x}_{t+1} = \boldsymbol{x}_t + \boldsymbol{f}(\boldsymbol{x}_t; \boldsymbol{A}) + \boldsymbol{\varepsilon}(\boldsymbol{x}_t, \boldsymbol{h}_t; \boldsymbol{\eta}) \tag{14a}$$

$$\boldsymbol{h}_{t+1} = \boldsymbol{h}_t + \sum_{l=1}^{L} (\boldsymbol{B}_l \boldsymbol{x}_{t+1} - \boldsymbol{C}_l \boldsymbol{h}_t) \tag{14b}$$

*where the input is given by* $\boldsymbol{x}_0 = \boldsymbol{\chi}_{\mathcal{S}}$ *and* $\boldsymbol{h}_0 = \boldsymbol{0}$.

The proof is given in Appendix B.3. The matrices $\boldsymbol{B}_l$ and $\boldsymbol{C}_l$ in (14b) correspond to the weights on $\boldsymbol{x}_{t+1}$ and $\boldsymbol{h}_t$ to form the *linear* transformation, and the neural network $\boldsymbol{\varepsilon}$ wraps up $(\boldsymbol{x}_t, \boldsymbol{h}_t)$ to approximate the *nonlinear* effect of the memory term in (10).

We here consider a more general convolution kernel $\boldsymbol{K}(\cdot; \boldsymbol{w})$ than the exponential kernel. Note that, in practice, the convolution weight $\boldsymbol{K}$ on older state $\boldsymbol{x}$ in (10) rapidly diminishes, and hence the memory kernel $\boldsymbol{K}$ can be well approximated with a truncated history of finite length $\tau > 0$, or $\tau \in \mathbb{N}$ after discretization. Hence, we substitute (14b) by

$$\boldsymbol{h}_t = \boldsymbol{K}^{\boldsymbol{w}} \boldsymbol{m}_t \quad \text{where} \quad \boldsymbol{K}^{\boldsymbol{w}} = [\boldsymbol{K}_0^{\boldsymbol{w}}, \ldots, \boldsymbol{K}_\tau^{\boldsymbol{w}}] \quad \text{and} \quad \boldsymbol{m}_t = [\boldsymbol{x}_t; \ldots; \boldsymbol{x}_{t-\tau}]. \tag{15}$$

Then we formulate the evolution of the augmented state $\boldsymbol{m}_t$ defined in (15) and follow (14a) to obtain a single evolution of $\boldsymbol{m}_t$ for $t = 0, \ldots, T-1$:

$$\boldsymbol{m}_{t+1} = \boldsymbol{g}(\boldsymbol{m}_t; \boldsymbol{\theta}), \quad \text{where} \quad \boldsymbol{g}(\boldsymbol{m}; \boldsymbol{\theta}) := [\boldsymbol{J}_0 \boldsymbol{m} + \tilde{\boldsymbol{f}}(\boldsymbol{J}_0 \boldsymbol{m}, \boldsymbol{K}^{\boldsymbol{w}} \boldsymbol{m}; \boldsymbol{\theta}); \boldsymbol{J}_0 \boldsymbol{m}; \ldots; \boldsymbol{J}_{\tau-1} \boldsymbol{m}] \tag{16}$$

and $\boldsymbol{J}_s := [\cdots, \boldsymbol{I}, \cdots] \in \mathbb{R}^{n \times (\tau+1)n}$ has identity $\boldsymbol{I}$ as the $(s+1)$th block and $\boldsymbol{0}$ elsewhere (thus $\boldsymbol{J}_s \boldsymbol{m}_t$ extracts the $(s+1)$th block $\boldsymbol{x}_{t-s}$ of $\boldsymbol{m}_t$) for $s = 0, \ldots, \tau-1$. If (14b) is considered, a simpler augmented state $\boldsymbol{m}_t = [\boldsymbol{x}_t; \boldsymbol{h}_t]$ can be formed similarly; we omit the details here. We will use the dynamics (16) of the augmented state $\boldsymbol{m}_t$ in the training below.

**An optimal control formulation of parameter learning** Now we consider the training of the network parameters $\boldsymbol{\theta} = (\boldsymbol{A}, \boldsymbol{\eta}, \boldsymbol{w})$ of (16) using cascade data $\mathcal{D}$. Given a sample cascade $\hat{\boldsymbol{x}} = (\mathcal{S}, \boldsymbol{\tau})$ from $\mathcal{D}$, we can observe its value in $\{0, 1\}^n$ at each of the time points $t = 1, \ldots, T$ and obtain the corresponding infection states, i.e., $\hat{\boldsymbol{x}} = \{\hat{\boldsymbol{x}}_t \in \{0, 1\}^n : t \in [T]\}$ (see Section 2). Maximizing the log-likelihood of $\hat{\boldsymbol{x}}$ for the dynamics $\boldsymbol{x}_t = \boldsymbol{x}_t(\boldsymbol{\theta}) \in [0, 1]^n$ induced by $\boldsymbol{\theta}$ is equivalent to minimizing the loss function $\ell(\boldsymbol{x}, \hat{\boldsymbol{x}})$:

$$\ell(\boldsymbol{x}, \hat{\boldsymbol{x}}) = \sum_{t=1}^{T} \hat{\boldsymbol{x}}_t \cdot \log \boldsymbol{x}_t + (\boldsymbol{1} - \hat{\boldsymbol{x}}_t) \cdot \log(\boldsymbol{1} - \boldsymbol{x}_t), \tag{17}$$

where the logarithm is taken componentwisely. We can add a regularization term $r(\boldsymbol{\theta})$ to (17) to impose prior knowledge or constraint on $\boldsymbol{\theta}$. In particular, if $\mathcal{E}$ is known, we can enforce a constraint such that $\boldsymbol{A}$ must be supported on $\mathcal{E}$ only. Otherwise, we can add $\|\boldsymbol{A}\|_1$ or $\|\boldsymbol{A}\|_0$ (the $l_1$ or $l_0$ norm of the vectorized $\boldsymbol{A}$) if $\mathcal{E}$ is expected to be sparse. In general, $\boldsymbol{A}$ can be interpreted as the convolution to be learned from a graph convolution network (GCN) [26, 53]. The support and magnitude of $\boldsymbol{A}$ imply the network structure and strength of interaction between nodes, respectively. We provide more details of our numerical implementation in Section 4 and Appendix D.1.

The optimal parameter $\boldsymbol{\theta}$ can be obtained by minimizing the loss function in (17) subject to the NMF dynamics (16). This procedure can also be cast as an optimal control problem to find $\boldsymbol{\theta}$ that steers $\boldsymbol{m}_t$ to fit data $\mathcal{D}$ through the NMF in (16):

$$\min_{\boldsymbol{\theta}} \quad \mathcal{J}(\boldsymbol{\theta}) := (1/K) \cdot \sum_{k=1}^{K} \ell(\boldsymbol{x}^{(k)}, \hat{\boldsymbol{x}}^{(k)}) + r(\boldsymbol{\theta}) \tag{18a}$$

$$\text{s.t.} \quad \boldsymbol{m}_{t+1}^{(k)} = \boldsymbol{g}(\boldsymbol{m}_t^{(k)}; \boldsymbol{\theta}), \quad \boldsymbol{m}_0^{(k)} = [\boldsymbol{\chi}_{\mathcal{S}_k}, \boldsymbol{0}, \ldots, \boldsymbol{0}], \quad t \in [T] - 1, \ k \in [K], \tag{18b}$$

where $\boldsymbol{x}_t^{(k)} = \boldsymbol{J}_0 \boldsymbol{m}_t^{(k)}$ for all $t$ and $k$. The problem of optimal control has been well studied in both continuous and discrete cases in the past decades [2]. In particular, the discrete optimal control

with nonlinear difference equations and the associated maximum principle have been extensively exploited. Recently, an optimal control viewpoint of deep learning has been proposed [32]—the network parameters of a neural network play the role of control variable in a discretized differential equation, and the training of these parameters for the network output to minimize the loss function can be viewed as finding the optimal control to minimize the objective function at the terminal state.

The Pontryagin's Maximum Principle (PMP) provides an important optimality condition of the optimal control [2, 32]. In standard optimal control, the control variable can be chosen freely in the allowed set at any given time $t$, which is a key in the proof of PMP. However, the NMF dynamics derived in (13) or (14) require a time invariant control $\boldsymbol{\theta}$ throughout. This is necessary since $\boldsymbol{\theta}$ corresponds to the network parameter and needs to be shared across different layers of the RNN, either from the linear kernel case with state $[\boldsymbol{x}; \boldsymbol{h}]$ in (14) or the general case with state $\boldsymbol{m}$ in (16). Therefore, we need to modify the original PMP and the optimality condition for our NMF formulation. To this end, consider the *Hamiltonian* function

$$H(\boldsymbol{m}, \boldsymbol{p}; \boldsymbol{\theta}) = \boldsymbol{p} \cdot \boldsymbol{g}(\boldsymbol{m}; \boldsymbol{\theta}) - \tfrac{1}{T} r(\boldsymbol{\theta}), \tag{19}$$

and define the *total Hamiltonian* of the system (14) as $\sum_{t=0}^{T-1} H(\boldsymbol{m}_t, \boldsymbol{p}_{t+1}; \boldsymbol{\theta})$. Then we can show that the optimal solution $\boldsymbol{\theta}^*$ is a time invariant control satisfying a *modified* PMP as follows.

**Theorem 4.** *Let $\boldsymbol{x}^*$ be the optimally controlled state process by $\boldsymbol{\theta}^*$, then there exists a co-state (adjoint) $\boldsymbol{p}^*$ which satisfies the backward differential equation*

$$\boldsymbol{m}_{t+1}^* = \boldsymbol{g}(\boldsymbol{m}_t^*; \boldsymbol{\theta}^*), \qquad \boldsymbol{m}_0^* = [\boldsymbol{\chi}_{\mathcal{S}_k}; \boldsymbol{0}; \ldots; \boldsymbol{0}], \quad t = 0, \ldots, T-1, \tag{20a}$$

$$\boldsymbol{p}_t^* = \boldsymbol{p}_{t+1}^* \cdot \nabla_{\boldsymbol{m}} \boldsymbol{g}(\boldsymbol{m}_t^*; \boldsymbol{\theta}^*), \quad \boldsymbol{p}_T^* = -\nabla_{\boldsymbol{m}_T} \ell, \quad t = T-1, \ldots, 0. \tag{20b}$$

*Moreover, the optimal $\boldsymbol{\theta}^*$ maximizes the total Hamiltonian: for any $\boldsymbol{\theta}$, there is*

$$\sum_{t=0}^{T-1} H(\boldsymbol{m}_t^*, \boldsymbol{p}_{t+1}^*; \boldsymbol{\theta}^*) \geq \sum_{t=0}^{T-1} H(\boldsymbol{m}_t^*, \boldsymbol{p}_{t+1}^*; \boldsymbol{\theta}). \tag{21}$$

*In addition, for any given $\boldsymbol{\theta}$, there is $\nabla_{\boldsymbol{\theta}} \mathcal{J}(\boldsymbol{\theta}) = -\sum_{t=0}^{T-1} \partial_{\boldsymbol{\theta}} H(\boldsymbol{m}_t^{\boldsymbol{\theta}}, \boldsymbol{p}_{t+1}^{\boldsymbol{\theta}}; \boldsymbol{\theta})$, where $\{\boldsymbol{m}_t^{\boldsymbol{\theta}}, \boldsymbol{p}_t^{\boldsymbol{\theta}} : 0 \leq t \leq T\}$ are obtained by the forward and backward passes (20a)-(20b) with $\boldsymbol{\theta}$.*

The proof is given in Appendix B.4. We introduced the *total Hamiltonian* $\sum_{t=0}^{T-1} H(\boldsymbol{m}_t, \boldsymbol{p}_{t+1}; \boldsymbol{\theta})$ in Theorem 4 since the NMF dynamics (14) (or (16)) suggest a *time invariant* control $\boldsymbol{\theta}$ independent of $t$, which corresponds to $\boldsymbol{\theta}$ shared by all layers in an RNN. This is particularly important for time series analysis, where we perform regression on data observed within limited time window, but often want to use the learned parameters to predict events in distant future. Theorem 4 also implies that performing gradient descent to minimize $\mathcal{J}$ in (18a) with back-propagation is equivalent to maximizing the total Hamiltonian in light of (21).

Our numerical implementation of the proposed NMF is summarized in Algorithm 1. From training cascade data $\mathcal{D}$, NMF can learn the parameter $\boldsymbol{\theta} = (\boldsymbol{A}, \boldsymbol{\eta}, \boldsymbol{w})$. The support (indices of nonzero entries) of the matrix $\boldsymbol{A}$ reveals the edge $\mathcal{E}$ of the diffusion network $\mathcal{G} = (\mathcal{V}, \mathcal{E})$, and the values of $\boldsymbol{A}$ are the corresponding infection rates on the edges. In addition to the diffusion network parameters inferred by $\boldsymbol{A}$, we can also estimate (predict) the influence $\{\boldsymbol{x}_t : t \in [T]\}$ of any new source set $\boldsymbol{x}_0 \in \mathbb{R}^n$ by a forward pass of NMF dynamics (16) with the learned $\boldsymbol{\theta}$. Note that this forward pass can be computed on the fly, which is critical to those downstream applications (such as influence maximization) that call influence estimation as a subroutine repeatedly during the computations.

## 4 Numerical Experiments

**Infection probability and influence function estimation** We first test NMF on a set of synthetic networks that mimic the structure of real-world diffusion network. Two types of the Kronecker network model [27] are used: hierarchical (Hier) [8] and core-periphery (Core) [28] networks with parameter matrices [0.9,0.1;0.1,0.9] and [0.9,0.5;0.5,0.3], respectively. For each type of network model, we generate 5 networks consisting of 128 nodes and 512 edges. We simulate the diffusion where the infection times are modeled by exponential distribution (Exp) and Rayleigh distribution (Ray). For each distribution, we draw $\alpha_{ji}$ from Unif[0.1,1] to simulate the varying interactions between nodes. We generate training data consists of $K$=10,000 cascades, which is formed by 10 sample cascades for each of 1,000 source sets (a source set is generated by randomly selecting 1 to

---

**Algorithm 1** Neural mean-field (NMF) algorithm for network inference and influence estimation

---

**Input:** $\mathcal{D} = \{\mathcal{C}_k : k \in [K]\}$ where $\mathcal{C}_k = \{\hat{\boldsymbol{x}}^{(k)}(t) \in \{0,1\}^n : t = 0, 1, \ldots, T\}$.
**Initialization:** Parameter $\boldsymbol{\theta} = (\boldsymbol{A}, \boldsymbol{\eta}, \boldsymbol{w})$.
**for** $k = 1, \ldots, K$ **do**
    Sample a mini-batch $\hat{\mathcal{D}} \subset \mathcal{D}$ of cascades.
    Compute $\{\boldsymbol{m}_t : t \in [T]\}$ using (16) with $\boldsymbol{\theta}$ and $\boldsymbol{m}_0 = [\boldsymbol{\chi}_{\mathcal{S}}; \boldsymbol{0}]$ for each $\mathcal{C} \in \hat{\mathcal{D}}$. (Forward pass)
    Compute $\hat{\nabla}_{\boldsymbol{\theta}} \mathcal{J} = \sum_{\mathcal{C} \in \hat{\mathcal{D}}} \nabla_{\boldsymbol{\theta}} \ell(\boldsymbol{x}, \hat{\boldsymbol{x}})$ with $\ell$ in (17).              (Backward pass)
    Update parameter $\boldsymbol{\theta} \leftarrow \boldsymbol{\theta} - \tau \hat{\nabla}_{\boldsymbol{\theta}} \mathcal{J}$.
**end for**
**Output:** Network parameter $\boldsymbol{\theta}$.

---

10 nodes from the network). All networks and cascades are generated by SNAP [29]. Our numerical implementation of NMF is available at `https://github.com/ShushanHe/neural-mf`.

We compare NMF to two baseline methods: InfluLearner [12] which is a state-of-the-art method that learns the coverage function of each node for any fixed time, and a conditional LSTM (LSTM for short) [22], which are among the few existing methods capable of learning infection probabilities of individual nodes directly from cascade data as ours. For InfluLearner, we set 128 as the feature number for optimal accuracy as suggested in [12]. For LSTM, we use one LSTM block and a dense layer for each $t$. To evaluate accuracy, we compute the mean absolute error (MAE) of node infection probability and influence over 100 source sets for each time $t$. More details of the evaluation criteria are provided in Appendix D.1. The results are given in Figure 1, which shows the mean (center line) and standard deviation (shade) of the three methods. NMF generally has lowest MAE, except at some early stages where InfluLearner is better. Note that InfluLearner requires and benefits from the knowledge of the original source node for each infection in the cascade (provided in our training data), which is often unavailable in practice and not needed in our method.

We also tested NMF on a real dataset [54] from Sina Weibo social platform consisting of more that 1.78 million users and 308 million following relationships among them. Following the setting in [12], we select the most popular tweet to generate diffusion cascades from the past 1,000 tweets of each user. Then we recreate the cascades by only keeping nodes of the top 1,000 frequency in the pooled node set over all cascades. For testing, we uniformly generate 100 source sets of size 10 and use $t = 1, 2, \ldots, 10$ as the time steps for observation. Finally, we test 100 source sets and compare our model NMF with the InfluLearner and LSTM. The MAE of all methods are shown in Figure 2a which shows that NMF significantly outperforms LSTM and is similar to InfluLearner. However, unlike InfluLearner that requires re-training for *every* $t$ and is computationally expensive, NMF learns the evolution at all $t$ in a single sweep of training and is tens of time faster.

We also test robustness of NMF for varying network density $|\mathcal{E}|/n$. The MAE of influence and infection probabilty by NMF on a hierarchical network with $n = 128$ are shown in Figure 2c and 2b, respectively. NMF remains accurate for denser networks, which can be notoriously difficult for other methods such as InfluLearner.

**Network structure inference** The interpretable parameterization of NMF allows us to explicitly learn the weight matrix $\boldsymbol{A}$. In this test, we examine the quality of the learned $\boldsymbol{A}$. We set the recovered adjacency matrix $\mathcal{E}$ to the binary indicator matrix $\boldsymbol{A}^\top \geq \epsilon$, i.e., $(\mathcal{E})_{i,j} = 1$ if $(\boldsymbol{A})_{ji} \geq 0.01$. To evaluate the quality of $\mathcal{E}$ and $\boldsymbol{A}$, we use four metrics: precision (Prc), recall (Rcl), accuracy (Acc), and correlation (Cor), defined as follows,

$$\text{Prc}(\mathcal{E}, \mathcal{E}^*) = \frac{|\mathcal{E} \cap \mathcal{E}^*|}{|\mathcal{E}^*|}, \ \text{Rcl}(\mathcal{E}, \mathcal{E}^*) = \frac{|\mathcal{E} \cap \mathcal{E}^*|}{|\mathcal{E}|}, \ \text{Acc}(\mathcal{E}, \mathcal{E}^*) = 1 - \frac{|\mathcal{E} - \mathcal{E}^*|}{|\mathcal{E}| + |\mathcal{E}^*|}, \ \text{Cor}(A, A^*) = \frac{\text{tr}(A^\top A^*)}{\|A\|_F \|A^*\|_F}.$$

where $\mathcal{E}^*$ and $\boldsymbol{A}^*$ are their true values, respectively. In Acc, the edge set $\mathcal{E}$ is also interpreted as a matrix, and $|\mathcal{E}|$ counts the number of nonzeros in $\mathcal{E}$. In Cor, $\|A\|_F^2 = \text{tr}(A^\top A)$ is the Frobenius norm of the matrix $A$. Prc is the ratio of edges in $\mathcal{E}^*$ that are recovered in $\mathcal{E}$. Rcl is the ratio of correctly recovered edges in $\mathcal{E}$. Acc indicates the ratio of the number of common edges shared by $\mathcal{E}$ and $\mathcal{E}^*$ against the total number of edges in them. Cor measures similarity between $A$ and $A^*$ by taking their values into consideration. All metrics are bounded between $[0, 1]$, and higher value indicates better result. For comparison, we also test NETRATE [16] to the cascade data and learn $\boldsymbol{A}$ with Rayleigh distribution. Evaluation by four metrics are shown in Table 1, which indicates that

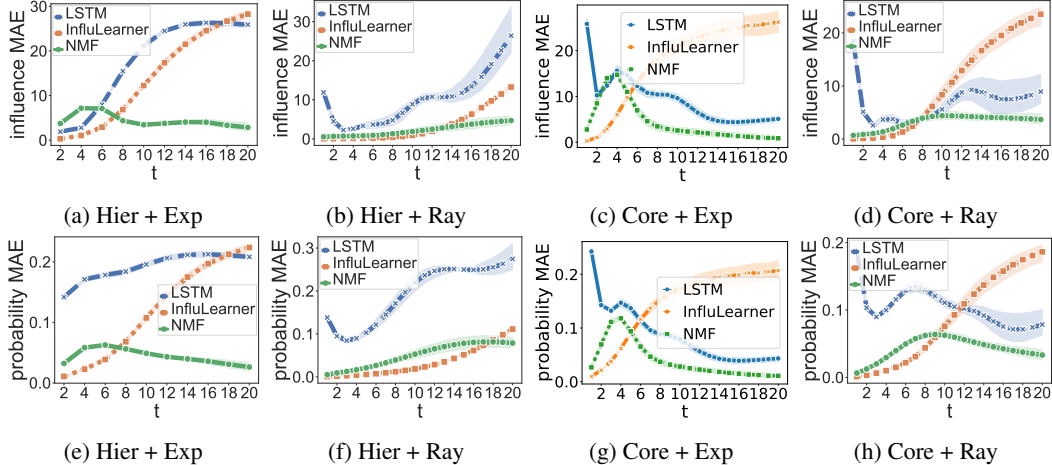

(a) Hier + Exp     (b) Hier + Ray     (c) Core + Exp     (d) Core + Ray

(e) Hier + Exp     (f) Hier + Ray     (g) Core + Exp     (h) Core + Ray

Figure 1: MAE of influence (top row) and node infection probability (bottom row) by LSTM, InfluLearner, and NMF on different combinations of Hierarchical (Hier) and Core-periphery (Core) networks, and exponential (Exp) and Rayleigh (Ray) diffusion models. Mean (centerline) and standard deviation (shade) over 100 tests are shown.

Table 1: Performance of structure inference using NETRATE and the proposed NMF on Random, Hierarchical, and Core-periphery networks with Rayleigh distribution as the diffusion time model on edges. Quality of the learned edge set $\mathcal{E}$ and distribution parameter $\boldsymbol{A}$ are measured by precision (Prc), recall (Rcl), accuracy (Acc), and correlation (Cor). Higher value indicates better quality.

| Network | Method | Prc | Rcl | Acc | Cor |
|---|---|---|---|---|---|
| Random | NETRATE | 0.481 | 0.399 | 0.434 | 0.465 |
|  | NMF | **0.858** | **0.954** | **0.903** | **0.950** |
| Hierarchical | NETRATE | 0.659 | 0.429 | 0.519 | 0.464 |
|  | NMF | **0.826** | **0.978** | **0.893** | **0.938** |
| Core-periphery | NETRATE | 0.150 | 0.220 | 0.178 | 0.143 |
|  | NMF | **0.709** | **0.865** | **0.779** | **0.931** |

NMF outperforms NETRATE in all metrics. Note that NMF learns $\boldsymbol{A}$ along with the NMF dynamics for infection probability estimation in its training, whereas NETRATE can only learn the matrix $\boldsymbol{A}$.

**Influence maximization**    We use NMF as an influence estimation subroutine in a classical greedy algorithm [38] (NMF+Greedy), and compare with a state-of-the-art method DIFFCELF[42] for influence maximization (IM). Like NMF+Greedy, DIFFCELF also only requires infection time features, but not network structures as in most existing methods. We generate 1000 cascades with unique source (as required by DIFFCELF but not ours) on a hierarchical network of 128 nodes and 512 edges, and use exponential distribution for the transmission function with $\boldsymbol{A}$ generated from Unif[1,10]. Time window is $T = 20$. For each source set size $n_0 = 1, \ldots, 10$, NMF+Greedy and DIFFCELF are applied to identify the optimal source sets, whose influence are computed by averaging 10,000 MC simulated cascades. Figure 2d shows that the source sets obtained by NMF+Greedy generates greater influence than DIFFCELF consistently for every source size $n_0$.

## 5   Related Work

Sampling-based influence estimation methods have been considered for discrete-time and continuous-time diffusion models. Discrete-time models assume node infections only occur at discrete time points. Under this setting, the Independent Cascade (IC) model [24] is considered and a method with provable performance guarantee is developed for single source which iterates over a sequence of guesses of the true influence until the verifier accepts in [34]. To resolve the inefficiency of Monte

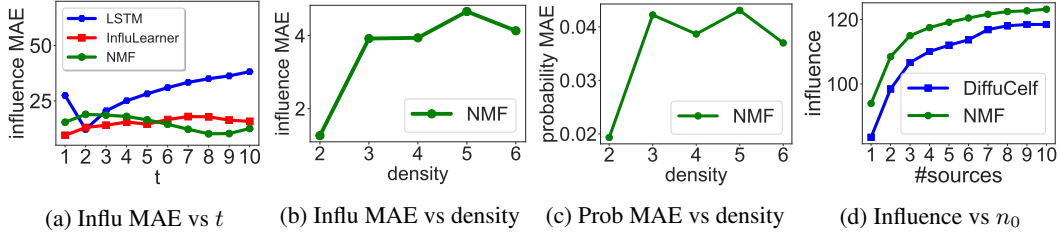

(a) Influ MAE vs $t$      (b) Influ MAE vs density    (c) Prob MAE vs density      (d) Influence vs $n_0$

Figure 2: (a) MAE of influence estimated by LSTM, InfluLearner on Weibo data; (b)–(c) MAE of influence and infection probability of NMF for different network densities; (d) Influence of source sets selected by DIFFUCELF and NMF+Greedy for $n_0 = 1, \ldots, 10$.

Carlo simulations, the reverse influence sampling (RIS) sketching method [5] is adopted in [41]. Moreover, instead of using the full network structure, sketch-based approaches only characterize propagation instances for influence computation, such as the method in [10], which considers per-node summary structures defined by the bottom-$k$ min-bash [9] sketch of the combined reachability set. In contrast to discrete-time models, continuous-time diffusion models allow arbitrary event occurrence times and hence are more accurate in modeling real-world diffusion processes. In Continuous-time Independent Cascade (CIC) models, influence estimation can be reformulated as the problem of finding the least label list which contains information about the distance to the smallest reachable labels from the source [13, 20]. Compared to methods using a fixed number of samples, a more scalable approximation scheme with a built-in block is developed to minimize the number of samples needed for the desired accuracy [40].

The aforementioned methods require knowledge of cascade traces [10] or the diffusion networks (review of related work on network structure inference is provided in Appendix C), such as node connectivity and node-to-node infection rates, as well as various assumptions on the diffusion of interests. However, such knowledge about the diffusion networks may not be available in practice, and the assumptions on the propagation or data formation are often application-specific and do not hold in most other problems. InfluLearner [12] is a state-of-the-art method that does not require knowledge of the underlying diffusion network. InfluLearner estimates the influence directly from cascades data in the CIC models by learning the influence function with a parameterization of the coverage functions using random basis functions. However, the random basis function suggested by [12] requires knowledge of the original source node for every infection, which can be difficult or impossible to be tracked in real-world applications.

In recent years, deep learning techniques have been employed to improve the scalability of influence estimation on large networks. In particular, convolutional neural networks (CNNs) and attention mechanism are incorporated with both network structures and user specific features to learn users' latent feature representation in [44]. By piping represented cascade graphs through a gated recurrent unit (GRU), the future incremental influence of a cascade can be predicted [31]. RNNs and CNNs are also applied to capture the temporal relationships on the user-generated contents networks (e.g., views, likes, comments, reposts) and extract more powerful features in [55]. In methods based on graph structures, graph neural networks (GNNs) and graph convolution networks (GCNs) are widely applied. In particular, two coupled GNNs are used to capture the interplay between node activation states and the influence spread [6], while GCNs integrated with teleport probability from the domain of page rank in [30] enhanced the performance of method in [44]. However, these methods depend critically on the structure or content features of cascades which may not be available in practice.

## 6 Conclusion

We proposed a novel framework using neural mean-field dynamics for inference and estimation on diffusion networks. Our new framework is derived from the Mori-Zwanzig formalism to obtain exact evolution of node infection probabilities. The memory term of the evolution can be approximated by convolutions, which renders the system as a delay differential equation and its time discretization reduces to a structured and interpretable RNN. Empirical study shows that our approach is versatile and robust to different variations of diffusion network models, and significantly outperforms existing approaches in accuracy and efficiency on both synthetic and real-world data sets.

## Broader Impact

This paper makes a significant contribution to the learning of structure and infection probabilities for diffusion networks, which is one of the central problems in the study of stochastic information propagation on large heterogeneous networks. The proposed neural mean-field (NMF) dynamics provide the first principled approach for inference and estimation problems using cascade data. NMF is shown to be a delay differential equation with proper approximation of memory integral using learnable time convolution operators, and the system reduces to a highly structured and interpretable recurrent neural network after time discretiztion. Potential applications include influence maximization, outbreak detection, and source identification.

## Acknowledgments and Disclosure of Funding

The work of SH and XY was supported in part by National Science Foundation under grants CMMI-1745382, DMS-1818886, and DMS-1925263. The work of HZ was supported in part by National Science Foundation IIS-1717916 and Shenzhen Research Institute of Big Data. He was on leave from College of Computing, Georgia Institute of Technology.

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
