[Supplementary Material]

# A An Illustrative Example of Mori-Zwanzig Formalism

The following problem is widely used as an introductory example of the Mori-Zwanzig (MZ) formalism [21, 51] for model order reduction: let $\boldsymbol{z} = [\boldsymbol{x}; \boldsymbol{y}] \in \mathbb{R}^N$ where $\boldsymbol{x} \in \mathbb{R}^n$, $\boldsymbol{y} \in \mathbb{R}^{N-n}$, and $n \ll N$. Consider the system of linear differential equations

$$\begin{cases} \boldsymbol{x}' &= \boldsymbol{A}_{11}\boldsymbol{x} + \boldsymbol{A}_{12}\boldsymbol{y}, \\ \boldsymbol{y}' &= \boldsymbol{A}_{21}\boldsymbol{x} + \boldsymbol{A}_{22}\boldsymbol{y}, \end{cases} \tag{22}$$

with initial values $\boldsymbol{x}(0) = \boldsymbol{x}_0$ and $\boldsymbol{y}(0) = \boldsymbol{y}_0$. Suppose we are interested in the time evolution of $\boldsymbol{x}(t)$, which depends on the joint effect of $\boldsymbol{x}$ and $\boldsymbol{y}$. However the computation of the complete system (22) is expensive and can be prohibitive for large $N$. The question is whether we can derive a reduced system only involving $\boldsymbol{x}$ from (22). To this end, we assume $\boldsymbol{x}$ is given, and solve for $\boldsymbol{y}$ from the $\boldsymbol{y}$-equation of (22) to obtain

$$\boldsymbol{y}(t) = e^{\boldsymbol{A}_{22}t}\boldsymbol{y}_0 + \int_0^t e^{\boldsymbol{A}_{22}(t-s)}\boldsymbol{A}_{22}x(s)\,\mathrm{d}s. \tag{23}$$

Then we plug this back into the $\boldsymbol{x}$-equation of (22) and obtain

$$\boldsymbol{x}'(t) = \boldsymbol{A}_{11}\boldsymbol{x}(t) + \boldsymbol{A}_{12}\int_0^t e^{\boldsymbol{A}_{22}(t-s)}\boldsymbol{A}_{22}\boldsymbol{x}(s)\,\mathrm{d}s + \boldsymbol{A}_{12}e^{\boldsymbol{A}_{22}t}\boldsymbol{y}_0, \tag{24}$$

which neglects the dependence on $\boldsymbol{y}(t)$ except for the initial value $\boldsymbol{y}_0$.

As shown in the example above, MZ formalism aims at reducing a high-dimensional system of $\boldsymbol{z}$ into a low-dimensional system of $\boldsymbol{x}$ (resolved variable) while maintaining the effect of $\boldsymbol{y}$ (unresolved variable). This is particularly useful if an exact solution of $\boldsymbol{z}$ is unnecessary to understand the dynamics of $\boldsymbol{x}$. Specialized derivations and subsequent approximation techniques can be implemented to obtain highly efficient numerical solutions for nonlinear systems.

# B Proofs

## B.1 Proof of Theorem 1

*Proof.* Let $\lambda_i^*(t)$ be the conditional intensity of node $i$ at time $t$, i.e., $\mathbb{E}[\mathrm{d}X_i(t)|\mathcal{H}(t)] = \lambda_i^*(t)\,\mathrm{d}t$. In the standard diffusion model, the conditional intensity $\lambda_i^*(t)$ of a healthy node $i$ (i.e., $X_i(t) = 0$) is determined by the total infection rate of its infected neighbors $j$ (i.e., $X_j(t) = 1$). That is,

$$\lambda_i^*(t) = \sum_j \alpha_{ji} X_j(t)(1 - X_i(t)). \tag{25}$$

By taking expectation $\mathbb{E}_{\mathcal{H}(t)}[\cdot]$ on both sides of (25), we obtain

$$\begin{aligned} \lambda_i(t) :&= \mathbb{E}_{\mathcal{H}(t)}[\lambda_i^*(t)] = \mathbb{E}_{\mathcal{H}(t)}\left[\alpha_{ji}X_j(t)(1 - X_i(t))\big|\mathcal{H}(t)\right] \\ &= \sum_j \alpha_{ji}(x_j - x_{ij}) = \sum_j \alpha_{ji}(x_j - y_{ij} - e_{ij}). \end{aligned} \tag{26}$$

On the other hand, there is

$$\lambda_i(t)\,\mathrm{d}t = \mathbb{E}_{\mathcal{H}(t)}[\lambda_i^*(t)]\,\mathrm{d}t = \mathbb{E}_{\mathcal{H}(t)}[\mathrm{d}X_i(t)|\mathcal{H}(t)] = \mathrm{d}\mathbb{E}_{\mathcal{H}(t)}[X_i(t)|\mathcal{H}(t)] = \mathrm{d}x_i. \tag{27}$$

Combining (26) and (27) yields

$$x_i' = \frac{\mathrm{d}x_i(t)}{\mathrm{d}t} = \sum_j \alpha_{ji}(x_j - y_{ij} - e_{ij}) = (\boldsymbol{A}\boldsymbol{x})_i - (\mathrm{diag}(\boldsymbol{x})\boldsymbol{A}\boldsymbol{x})_i - \sum_j \alpha_{ji}e_{ij}$$

for every $i \in [n]$, which verifies the $\boldsymbol{x}$ part of (6). Similarly, we can obtain

$$x_I' = \sum_{i \in I}\sum_{j \notin I} \alpha_{ji}(x_I - x_{I \cup \{j\}}) = \sum_{i \in I}\sum_{j \notin I} \alpha_{ji}(y_I + e_I - y_{I \cup \{j\}} - e_{I \cup \{j\}}). \tag{28}$$

Moreover, by taking derivative on both sides of $x_I(t) = y_I(t) + e_I(t)$, we obtain

$$x_I' = \sum_{i \in I} y_{I \setminus \{i\}} x_i' + e_I' = \sum_{i \in I} y_{I \setminus \{i\}} \sum_{j \neq i} \alpha_{ji}(x_j - x_i x_j - e_{ij}) + e_I'. \tag{29}$$

Combining (28) and (29) yields the $e$ part of (6).

It is clear that $\boldsymbol{x}_0 = \boldsymbol{\chi}_{\mathcal{S}}$. For every $I$, at time $t = 0$, there is $x_I(0) = \prod_{i \in I} X_i(0) = 1$ if $I \subset \mathcal{S}$ and 0 otherwise; and the same for $y_I(0)$. Hence $e_I(0) = x_I(0) - y_I(0) = 0$ for all $I$. Hence $\boldsymbol{z}_0 = [\boldsymbol{x}_0; \boldsymbol{e}_0] = [\boldsymbol{\chi}_{\mathcal{S}}; \boldsymbol{0}]$, which verifies the initial condition of (6). $\qquad \square$

## B.2  Proof of Theorem 2

*Proof.* Consider the system (6) over a finite time horizon $[0, T]$, which evolves on a smooth manifold $\Gamma \subset \mathbb{R}^N$. For any real-valued phase (observable) space function $g : \Gamma \to \mathbb{R}$, the nonlinear system (6) is equivalent to the linear partial differential equation, known as the Liouville equation:

$$\begin{cases} \partial_t u(t, \boldsymbol{z}) = \mathcal{L}[u](t, \boldsymbol{z}), \\ u(0, \boldsymbol{z}) = g(\boldsymbol{z}), \end{cases} \tag{30}$$

where the Liouville operator $\mathcal{L}[u] := \bar{\boldsymbol{f}}(\boldsymbol{z}) \cdot \nabla_{\boldsymbol{z}} u$. The equivalency is in the sense that the solution of (30) satisfies $u(t, \boldsymbol{z}_0) = g(\boldsymbol{z}(t; \boldsymbol{z}_0))$, where $\boldsymbol{z}(t; \boldsymbol{z}_0)$ is the solution to (6) with initial value $\boldsymbol{z}_0$.

Denote $e^{t\mathcal{L}}$ the Koopman operator associated with $\mathcal{L}$ such that $e^{t\mathcal{L}} g(\boldsymbol{z}_0) = g(\boldsymbol{z}(t))$ where $\boldsymbol{z}(t)$ is the solution of (6). Then $e^{t\mathcal{L}}$ satisfies the semi-group property, i.e.,

$$e^{t\mathcal{L}} g(\boldsymbol{z}) = g(e^{t\mathcal{L}} \boldsymbol{z}) \tag{31}$$

for all $g$. On the right hand side of (31), $\boldsymbol{z}$ can be interpreted as $\boldsymbol{z} = \boldsymbol{\iota}(\boldsymbol{z}) = [\iota_1(\boldsymbol{z}), \ldots, \iota_N(\boldsymbol{z})]$ where $\iota_j(\boldsymbol{z}) = z_j$ for all $j$.

Now consider the projection operator $\mathcal{P}$ as the truncation such that $\mathcal{P}g(\boldsymbol{z}) = \mathcal{P}g(\boldsymbol{x}, \boldsymbol{e}) = g(\boldsymbol{x}, 0)$ for any $\boldsymbol{z} = (\boldsymbol{x}, \boldsymbol{e})$, and its orthogonal complement as $\mathcal{Q} = I - \mathcal{P}$ where $I$ is the identity operator. Note that $\boldsymbol{z}'(t) = \frac{\mathrm{d}\boldsymbol{z}(t)}{\mathrm{d}t} = \frac{\partial}{\partial t} e^{t\mathcal{L}} \boldsymbol{z}_0$, and $\bar{\boldsymbol{f}}(\boldsymbol{z}(t)) = e^{t\mathcal{L}} \boldsymbol{f}(\boldsymbol{z}_0) = e^{t\mathcal{L}} \mathcal{L} \boldsymbol{z}_0$ since $\mathcal{L}\iota_j(\boldsymbol{z}) = \boldsymbol{f}_j(\boldsymbol{z})$ for all $\boldsymbol{z}$ and $j$. Therefore (6) implies that

$$\frac{\partial}{\partial t} e^{t\mathcal{L}} \boldsymbol{z}_0 = e^{t\mathcal{L}} \mathcal{L} \boldsymbol{z}_0 = e^{t\mathcal{L}} \mathcal{P} \mathcal{L} \boldsymbol{z}_0 + e^{t\mathcal{L}} \mathcal{Q} \mathcal{L} \boldsymbol{z}_0. \tag{32}$$

Note that the first term on the right hand side of (32) is

$$e^{t\mathcal{L}} \mathcal{P} \mathcal{L} \boldsymbol{z}_0 = \mathcal{P} \mathcal{L} e^{t\mathcal{L}} \boldsymbol{z}_0 = \mathcal{P} \mathcal{L} \boldsymbol{z}(t). \tag{33}$$

For the second term in (32), we recall that the well-known Dyson's identity for the Koopman operator $\mathcal{L}$ is given by

$$e^{t\mathcal{L}} = e^{t\mathcal{Q}\mathcal{L}} + \int_0^t e^{s\mathcal{L}} \mathcal{P} \mathcal{L} e^{(t-s)\mathcal{Q}\mathcal{L}} \, \mathrm{d}s. \tag{34}$$

Applying (34) to $\mathcal{Q}\mathcal{L}\boldsymbol{z}_0$ yields

$$e^{t\mathcal{L}} \mathcal{Q}\mathcal{L} \boldsymbol{z}_0 = e^{t\mathcal{Q}\mathcal{L}} \mathcal{Q}\mathcal{L} \boldsymbol{z}_0 + \int_0^t e^{s\mathcal{L}} \mathcal{P}\mathcal{L} e^{(t-s)\mathcal{Q}\mathcal{L}} \mathcal{Q}\mathcal{L} \boldsymbol{z}_0 \, \mathrm{d}s$$

$$= e^{t\mathcal{Q}\mathcal{L}} \mathcal{Q}\mathcal{L} \boldsymbol{z}_0 + \int_0^t \mathcal{P}\mathcal{L} e^{(t-s)\mathcal{Q}\mathcal{L}} \mathcal{Q}\mathcal{L} e^{s\mathcal{L}} \boldsymbol{z}_0 \, \mathrm{d}s \tag{35}$$

$$= e^{t\mathcal{Q}\mathcal{L}} \mathcal{Q}\mathcal{L} \boldsymbol{z}_0 + \int_0^t \mathcal{P}\mathcal{L} e^{(t-s)\mathcal{Q}\mathcal{L}} \mathcal{Q}\mathcal{L} \boldsymbol{z}(s) \, \mathrm{d}s.$$

Substituting (33) and (35) into (32), we obtain

$$\frac{\partial}{\partial t} e^{t\mathcal{L}} \boldsymbol{z}_0 = \mathcal{P}\mathcal{L}\boldsymbol{z}(t) + e^{t\mathcal{Q}\mathcal{L}} \mathcal{Q}\mathcal{L} \boldsymbol{z}_0 + \int_0^t \mathcal{P}\mathcal{L} e^{(t-s)\mathcal{Q}\mathcal{L}} \mathcal{Q}\mathcal{L}\boldsymbol{z}(s) \, \mathrm{d}s, \tag{36}$$

where we used the fact that $e^{t\mathcal{L}} \mathcal{P}\mathcal{L}\boldsymbol{z}_0 = \mathcal{P}\mathcal{L} e^{t\mathcal{L}} \boldsymbol{z}_0 = \mathcal{P}\mathcal{L}\boldsymbol{z}(t)$. Denote $\phi(t, \boldsymbol{z}) := e^{t\mathcal{L}} \mathcal{Q}\mathcal{L}\boldsymbol{z}$, then we simplify (36) into

$$\frac{\partial}{\partial t} e^{t\mathcal{L}} \boldsymbol{z}_0 = \mathcal{P}\mathcal{L}\boldsymbol{z}(t) + \phi(t, \boldsymbol{z}_0) + \int_0^t \boldsymbol{k}(t - s, \boldsymbol{z}(s)) \, \mathrm{d}s, \tag{37}$$

where $\boldsymbol{k}(t, \boldsymbol{z}) := \mathcal{P}\mathcal{L}\phi(t, \boldsymbol{z}) = \mathcal{P}\mathcal{L}e^{t\mathcal{L}}\mathcal{Q}\mathcal{L}\boldsymbol{z}$.

Now consider the evolution of $\phi(t, \boldsymbol{z})$, which is given by

$$\partial_t \phi(t, \boldsymbol{z}_0) = \mathcal{Q}\mathcal{L}\phi(t, \boldsymbol{z}_0), \tag{38}$$

with initial condition $\phi(0, \boldsymbol{z}_0) = \mathcal{Q}\mathcal{L}\boldsymbol{z}_0 = \mathcal{L}\boldsymbol{z}_0 - \mathcal{P}\mathcal{L}\boldsymbol{z}_0 = \bar{\boldsymbol{f}}(\boldsymbol{x}_0, \boldsymbol{e}_0) - \bar{\boldsymbol{f}}(\boldsymbol{x}_0, \boldsymbol{0}) = \boldsymbol{0}$ since $\boldsymbol{e}_0 = \boldsymbol{0}$. Applying $\mathcal{P}$ on both sides of (38) yields

$$\partial_t \mathcal{P}\phi(t, \boldsymbol{z}_0) = \mathcal{P}\mathcal{Q}\mathcal{L}\phi(t, \boldsymbol{z}_0) = \boldsymbol{0},$$

with initial $\mathcal{P}\phi(0, \boldsymbol{z}_0) = \boldsymbol{0}$. This implies that $\mathcal{P}\phi(t, \boldsymbol{z}_0) = \boldsymbol{0}$ for all $t$. Hence, applying $\mathcal{P}$ to both sides of (36) yields

$$\frac{\partial}{\partial t}\mathcal{P}\boldsymbol{z}(t) = \frac{\partial}{\partial t}\mathcal{P}e^{t\mathcal{L}}\boldsymbol{z}_0 = \mathcal{P}\mathcal{L}\boldsymbol{z}(t) + \int_0^t \mathcal{P}\boldsymbol{k}(t-s, \boldsymbol{z}(s)) \, \mathrm{d}s. \tag{39}$$

Restricting to the first $n$ components, $\mathcal{P}\boldsymbol{z}(t)$ reduces to $\boldsymbol{x}(t)$ and $\mathcal{P}\boldsymbol{k}(t-s, \boldsymbol{z}(s))$ reduces to $\boldsymbol{k}(t-s, \boldsymbol{x}(s))$. Recalling that $\mathcal{P}\mathcal{L}\boldsymbol{z}(t) = \mathcal{P}\bar{\boldsymbol{f}}(\boldsymbol{z}(t)) = \bar{\boldsymbol{f}}(\boldsymbol{x}(t), \boldsymbol{0}) = \boldsymbol{f}(\boldsymbol{x}(t))$ completes the proof. $\square$

## B.3 Proof of Theorem 3

*Proof.* From the definition of $\boldsymbol{h}(t)$ in (40), we obtain

$$\boldsymbol{h} = \int_0^t \boldsymbol{K}(t-s; \boldsymbol{w})\boldsymbol{x}(s) \, \mathrm{d}s = \int_{-\infty}^t \boldsymbol{K}(t-s; \boldsymbol{w})\boldsymbol{x}(s) \, \mathrm{d}s = \int_0^\infty \boldsymbol{K}(s; \boldsymbol{w})\boldsymbol{x}(t-s) \, \mathrm{d}s \tag{40}$$

where we used the fact that $\boldsymbol{x}(t) = 0$ for $t < 0$. Taking derivative on both sides of (40) yields

$$\boldsymbol{h}' = \int_0^\infty \boldsymbol{K}(s; \boldsymbol{w})\boldsymbol{x}'(t-s) \, \mathrm{d}s = \int_0^\infty \boldsymbol{K}(s; \boldsymbol{w})\tilde{\boldsymbol{f}}(\boldsymbol{x}(t-s), \boldsymbol{h}(t-s); \boldsymbol{A}, \boldsymbol{\eta}) \, \mathrm{d}s$$

$$= \int_{-\infty}^t \boldsymbol{K}(t-s; \boldsymbol{w})\tilde{\boldsymbol{f}}(\boldsymbol{x}(s), \boldsymbol{h}(s); \boldsymbol{A}, \boldsymbol{\eta}) \, \mathrm{d}s = \int_0^t \boldsymbol{K}(t-s; \boldsymbol{w})\tilde{\boldsymbol{f}}(\boldsymbol{x}(s), \boldsymbol{h}(s); \boldsymbol{A}, \boldsymbol{\eta}) \, \mathrm{d}s$$

where we used the fact that $\boldsymbol{x}'(t) = \tilde{\boldsymbol{f}}(\boldsymbol{x}(t), \boldsymbol{h}(t); \boldsymbol{A}, \boldsymbol{\eta}) = 0$ for $t < 0$ in the last equality.

If $\boldsymbol{K}(t; \boldsymbol{w}) = \sum_l \boldsymbol{B}_l e^{-\boldsymbol{C}_l t}$, then we can take derivative of (40) and obtain

$$\boldsymbol{h}'(t) = \sum_{l=1}^L \frac{\mathrm{d}}{\mathrm{d}t}\left(\int_{-\infty}^t \boldsymbol{B}_l e^{-\boldsymbol{C}_l t}\boldsymbol{x}(s) \, \mathrm{d}s\right) = \sum_{l=1}^L \left(\boldsymbol{B}_l \boldsymbol{x}(t) - \int_{-\infty}^t \boldsymbol{B}_l \boldsymbol{C}_l e^{-\boldsymbol{C}_l t}\boldsymbol{x}(s) \, \mathrm{d}s\right)$$

$$= \sum_{l=1}^L \left(\boldsymbol{B}_l \boldsymbol{x}(t) - \boldsymbol{C}_l \int_{-\infty}^t \boldsymbol{B}_l e^{-\boldsymbol{C}_l t}\boldsymbol{x}(s) \, \mathrm{d}s\right) = \sum_{l=1}^L (\boldsymbol{B}_l \boldsymbol{x}(t) - \boldsymbol{C}_l \boldsymbol{h}(t)).$$

Time discretization (14) can then be obtained by finite difference in time with normalized step size 1 and proper scaling of the network parameters $\boldsymbol{\theta}$. $\square$

## B.4 Proof of Theorem 4

*Proof.* We consider the augmented state $\boldsymbol{\xi}$ and nonlinear dynamics $\bar{\boldsymbol{g}}(\cdot; \boldsymbol{\theta})$ associated with $\boldsymbol{m}$ and $\boldsymbol{g}(\cdot; \boldsymbol{\theta})$, defined as follows:

$$\boldsymbol{\xi}_0 = \begin{bmatrix} \boldsymbol{m}_0 \\ \boldsymbol{0} \\ \vdots \\ \boldsymbol{0} \end{bmatrix}, \quad \boldsymbol{\xi}_1 = \bar{\boldsymbol{g}}(\boldsymbol{\xi}_0; \boldsymbol{\theta}) := \begin{bmatrix} \boldsymbol{g}^1(\boldsymbol{m}_0; \boldsymbol{\theta}) \\ \boldsymbol{g}^2(\boldsymbol{m}_0; \boldsymbol{\theta}) \\ \vdots \\ \boldsymbol{g}^T(\boldsymbol{m}_0; \boldsymbol{\theta}) \end{bmatrix} = \begin{bmatrix} \boldsymbol{m}_1 \\ \boldsymbol{m}_2 \\ \vdots \\ \boldsymbol{m}_T \end{bmatrix}, \tag{41}$$

where $\boldsymbol{g}^t$ stands for the composition of $\boldsymbol{g}(\cdot; \boldsymbol{\theta})$ for $t$ times.

Without overloading the notations, we reuse $\mathcal{J}$ and $\ell$ of the objective function (18a) and loss function (17) of $\boldsymbol{m}$ respectively for the augmented state $\boldsymbol{\xi}$. In addition, following [32], we further simpify the notation by combining the $K$ training data into a single variable $\hat{\boldsymbol{x}} := [\hat{\boldsymbol{x}}^{(1)}, \ldots, \hat{\boldsymbol{x}}^{(K)}]$; similar

for the state variable $\boldsymbol{x}$. In this case, the dynamics $\boldsymbol{g}$ is applied to each column of $\boldsymbol{x}$, and the loss function $\ell$ is to be interpreted as the average loss as in (17). Furthermore, we temporarily assume the regularization $r(\boldsymbol{\theta}) = 0$ as it is simple to append $\boldsymbol{\theta}$ to the state $\boldsymbol{\xi}$ and merge $r(\boldsymbol{\theta})$ into the loss function $\ell(\boldsymbol{\xi}, \hat{\boldsymbol{\xi}})$. Then the optimal control problem (18) is rewritten as

$$\min_{\boldsymbol{\theta}} \quad \mathcal{J}(\boldsymbol{\theta}) := \ell(\boldsymbol{\xi}, \hat{\boldsymbol{\xi}}) + r(\boldsymbol{\theta}) \tag{42a}$$

$$\text{s.t.} \quad \boldsymbol{\xi}_1 = \bar{\boldsymbol{g}}(\boldsymbol{\xi}_0; \boldsymbol{\theta}), \quad \boldsymbol{\xi}_0 = [\boldsymbol{m}_0; \boldsymbol{0}; \dots; \boldsymbol{0}]. \tag{42b}$$

Note that (42) is a one-step optimal control with $\bar{\boldsymbol{g}}(\cdot; \boldsymbol{\theta})$. Now by the discrete Pontryagin's Maximum Principle [2], for the state $\boldsymbol{\xi}^*$ optimally controlled by $\boldsymbol{\theta}^*$, there exists a co-state $\boldsymbol{\psi}^*$, such that $\boldsymbol{\xi}^*$ and $\boldsymbol{\psi}^*$ satisfy the following forward and backward equations for $\boldsymbol{\theta} = \boldsymbol{\theta}^*$:

$$\boldsymbol{\xi}_1^* = \bar{\boldsymbol{g}}(\boldsymbol{\xi}_0^*; \boldsymbol{\theta}^*), \qquad \boldsymbol{\xi}_0^* = [\boldsymbol{m}_0; \boldsymbol{0}; \dots; \boldsymbol{0}], \tag{43a}$$

$$\boldsymbol{\psi}_0^* = \boldsymbol{\psi}_1^* \cdot \nabla_{\boldsymbol{\xi}} \bar{\boldsymbol{g}}(\boldsymbol{\xi}_1^*; \boldsymbol{\theta}^*), \quad \boldsymbol{\psi}_1^* = -\nabla_{\boldsymbol{\xi}} \ell(\boldsymbol{\xi}_1^*, \hat{\boldsymbol{\xi}}), \tag{43b}$$

where

$$\boldsymbol{\xi}_1^* = [\boldsymbol{m}_1^*; \dots; \boldsymbol{m}_T^*] \quad \text{and} \quad \boldsymbol{\psi}_1^* = [\partial_{\boldsymbol{m}_1} \ell(\boldsymbol{\xi}_1^*, \hat{\boldsymbol{\xi}}); \dots; \partial_{\boldsymbol{m}_T} \ell(\boldsymbol{\xi}_1^*, \hat{\boldsymbol{\xi}})] = [\boldsymbol{p}_1^*; \dots; \boldsymbol{p}_T^*]. \tag{44}$$

In addition, $\boldsymbol{\theta}^*$ maximizes the Hamiltonian $\mathcal{H}$ associated with (43):

$$\mathcal{H}(\boldsymbol{\xi}^*, \boldsymbol{\psi}^*; \boldsymbol{\theta}^*) \geq \mathcal{H}(\boldsymbol{\xi}^*, \boldsymbol{\psi}^*; \boldsymbol{\theta}), \quad \forall \boldsymbol{\theta}, \quad \text{where} \quad \mathcal{H}(\boldsymbol{\xi}, \boldsymbol{\psi}; \boldsymbol{\theta}) := \boldsymbol{\psi}_1 \cdot \bar{\boldsymbol{g}}(\boldsymbol{\xi}_0; \boldsymbol{\theta}) - r(\boldsymbol{\theta}). \tag{45}$$

Combining (44), (45), and the definition of $H$ in (19) yields the maximization of total Hamiltonian at the optimal control $\boldsymbol{\theta}^*$:

$$\sum_{t=0}^{T-1} H(\boldsymbol{m}_t^*, \boldsymbol{p}_{t+1}^*; \boldsymbol{\theta}^*) \geq \sum_{t=0}^{T-1} H(\boldsymbol{m}_t^*, \boldsymbol{p}_{t+1}^*; \boldsymbol{\theta}), \quad \forall \boldsymbol{\theta}.$$

For any control $\boldsymbol{\theta}$ and its state and co-state variables $\boldsymbol{\xi}^{\boldsymbol{\theta}}$ and $\boldsymbol{\psi}^{\boldsymbol{\theta}}$ following (43) with $\boldsymbol{\theta}$ (also corresponding to $\boldsymbol{m}_t^{\boldsymbol{\theta}}$ and $\boldsymbol{p}_t^{\boldsymbol{\theta}}$ for $t = 0, \dots, T$), we have

$$\begin{aligned}
\nabla_{\boldsymbol{\theta}} \mathcal{J}(\boldsymbol{\theta}) &= \nabla_{\boldsymbol{\xi}} \ell(\boldsymbol{\xi}_1^{\boldsymbol{\theta}}, \hat{\boldsymbol{\xi}}) \cdot \nabla_{\boldsymbol{\theta}} \boldsymbol{\xi}_1^{\boldsymbol{\theta}} + \nabla_{\boldsymbol{\theta}} r(\boldsymbol{\theta}) \\
&= [\partial_{\boldsymbol{m}_1} \ell(\boldsymbol{\xi}_1^{\boldsymbol{\theta}}, \hat{\boldsymbol{\xi}}); \dots; \partial_{\boldsymbol{m}_T} \ell(\boldsymbol{\xi}_1^{\boldsymbol{\theta}}, \hat{\boldsymbol{\xi}})] \cdot [\partial_{\boldsymbol{\theta}} \boldsymbol{g}(\boldsymbol{m}_0^{\boldsymbol{\theta}}; \boldsymbol{\theta}); \dots; \partial_{\boldsymbol{\theta}} \boldsymbol{g}(\boldsymbol{m}_{T-1}^{\boldsymbol{\theta}}; \boldsymbol{\theta})] + \nabla_{\boldsymbol{\theta}} r(\boldsymbol{\theta}) \\
&= -\sum_{t=1}^{T} \left( \boldsymbol{p}_t^{\boldsymbol{\theta}} \cdot \partial_{\boldsymbol{\theta}} \boldsymbol{g}(\boldsymbol{m}_t^{\boldsymbol{\theta}}; \boldsymbol{\theta}) + \frac{1}{T} \nabla_{\boldsymbol{\theta}} r(\boldsymbol{\theta}) \right) \\
&= -\sum_{t=1}^{T} \partial_{\boldsymbol{\theta}} H(\boldsymbol{m}_t^{\boldsymbol{\theta}}, \boldsymbol{p}_{t+1}^{\boldsymbol{\theta}}; \boldsymbol{\theta}),
\end{aligned}$$

which completes the proof. $\qquad\square$

## C Additional Related Work

**Network structure inference** Inference of diffusion network structure is an important problem closely related to influence estimation. In particular, if the network structure and infections rates are unknown, one often needs to first infer such information from a training dataset of sampled cascades, each of which tracks a series of infection times and locations on the network. Existing methods have been proposed to infer network connectivity [18, 45, 33, 14] and also the infection rates between nodes [37, 17, 19]. Submodular optimization is applied to infer network connectivity [18, 45, 33] by considering the most probable [18] or all [45, 33] directed trees supported by each cascade. One of the early works that incorporate spatio-temporal factors into network inference is introduced in [33]. Utilizing convex optimization, transmission functions [14], the prior probability [37], and the transmission rate [17] over edges are inferred from cascades. In addition to static networks, the infection rates are considered but also in the unobserved dynamic network changing over time [19]. Besides cascades, other features of dynamical processes on networks have been used to infer the diffusion network structures. To avoid using predefined transmission models, the statistical difference of the infection time intervals between nodes in the same cascade versus those not in any cascade was considered in [46]. A given time series of the epidemic prevalence, i.e., the average fraction of infected nodes was applied to discover the underlying network. The recurrent cascading behavior is also explained by integrating a feature vector describing the additional features [50]. A graph signal processing (GSP) approach is developed to infer graph structure from dynamics on networks [35, 11].

## D    Experiment Supplements

### D.1    Implementation details

In our NMF implementation, we use a standard LSTM architecture and 3 dense layers for the RNN $\varepsilon$ at each time $t$. Regularization terms using $l_1$-norm of all parameters are added to the loss function to promote their sparsity and robustness. Specifically, we use 0.001 to weight $\boldsymbol{A}$ and 0.0001 to all other trainable parameters, respectively. The NMF networks are trained and tested in TensorFlow [1] using Adam optimizer with default parameters (lr=0.001, $\beta_1$=0.9, $\beta_2$=0.999, $\epsilon$=1e-8) on a Linux workstation with Intel 8-Core Turbo 5GHz CPU, 64GB of memory, and an Nvidia RTX 2080Ti GPU. The LSTM model is trained and tested in the same setting as NMF except a fixed regularization weight 0.001 for all trainable parameters. InfluLearner is trained in Matlab, and the number of features is set to 128. All experiments are performed on the same machine. Given ground truth node infection probability $\boldsymbol{x}^*$, the Mean Absolute Error (MAE) of influence (Inf) and infection probability (Prob) of estimated $\boldsymbol{x}$ are defined by $|\boldsymbol{1} \cdot (\boldsymbol{x}_t - \boldsymbol{x}_t^*)|$ and $\|\boldsymbol{x}_t - \boldsymbol{x}_t^*\|_1/n$ for every $t$, respectively.

### D.2    Inference of node interdependencies

Due to its highly interpretable structure, NMF can also learn the node inter-dependencies through $\boldsymbol{A}$. In addition to the quantitative evaluations provided in Section 4, we show the visual appearance of $\boldsymbol{A}$ inferred by NMF in Figure 3. The ground truth $\boldsymbol{A}^*$ and $\boldsymbol{A}$ inferred by NETRATE are also provided for comparison. As we can see, $\boldsymbol{A}$ inferred by NMF is much more faithful to $\boldsymbol{A}^*$ than that by NETRATE. Note that NETRATE requires knowledge of specific diffusion model type (Rayleigh in this test) whereas NMF does not. This result shows that NMF is versatile and robust when only cascade data are available.

(a) True                       (b) NETRATE                       (c) NMF

Figure 3: Ground truth $\boldsymbol{A}^*$ (left) and $\boldsymbol{A}$ inferred NETRATE (middle) and NMF (right) under the same color scale using cascaded data from a Hierarchical network with Rayleigh diffusion model.

### D.3    Accuracy and Scalability

**Accuracy for networks of increasing sizes**    We test NMF on networks of increasing sizes up to $n$=2,048 with $|\mathcal{E}| = 2n$ for each $n$ using Hierarchical network and exponential diffusion model on cascade data containing 10,000 cascades. We also generate 100 extra cascades with 20%-validation and 80%-test. Figure 4 (a)–(b) shows the MAE of influence (Inf) and infection probability (Prob) estimated by NMF versus time for varying $n$, which indicate that the error remains low for large networks.

**Scalability**    We compare NMF to InfluLearner in terms of runtime for the influence estimation. For InfluLearner, we draw 200 features. For NMF, the batch size of training cascade data is set to 50 for the network with more than 2,048 nodes, and is 100 for smaller networks. The training is terminated when the average MAE of infection probability on validation data does not decrease for 20 epochs. Figure 4 (c) shows the comparison on runtime (in seconds) of training as we increase the network size $n$ in InfluLearner and NMF. Note that the original implementation of InfluLearner [12] is in

Matlab and the computational time increases drastically in network density, whereas our method retains similar runtime regardless of network density.

(a) Inf MAE vs t      (b) Prob MAE vs t      (c) Runtime vs # nodes

Figure 4: (a)–(b) MAE of influence (Inf) and infection probability (Prob) estimated by NMF for Hierarchical networks with increasing network sizes from 256 to 2048. (c) runtime (in seconds) for training versus network sizes in log-log scale.

### D.4 Additional results of infection probability estimation

We test a total of 9 combinations of network structures and diffusion models. Specifically, we generate Hierarchical (Hier), Core-periphery (Core), and Random (Rand) networks, and use Exponential (Exp), Rayleigh (Ray) and Weibull (Wbl) diffusion models on each of these networks. All scale and shape parameters are drawn from $\text{Unif}[0.1, 1]$ and $\text{Unif}[1, 10]$, respectively. Here we stretch NMF and apply to Weibull diffusion model even it has two parameters for each edge. The experiment setting and evaluation metrics are the same as in Section 4. The MAE of influence and node infection probabilities are shown in Figure 5, which shows that NMF consistently performs well with low estimation error after trained by cascade data. Again, it is worth noting that InfluLearner requires the identity of source node for every infection in the entire cascade during training, which is generally not available in practice nor needed in NMF.

Figure 5: MAE of influence (top) and node infection probability (bottom) by LSTM, InfluLearner, and NMF on each of the 9 different combinations of Hierarchical (Hier), Core-periphery (Core) and Random (Rand) networks, and exponential (Exp), Rayleigh (Ray) and Weibull (Wbl) diffusion models. Mean (centerline) and standard deviation (shade) over 100 tests are shown.