[Reviews · NeurIPS 2020]

Review 1

Summary and Contributions: The authors focus on the problem of inferring structural and dynamical properties of diffusion processes on graphs from a set of observations. To address this problem they developed a new methodology based on a rigorous analytical characterization of diffusion on graphs. They demonstrate on synthetic and real-world data that their newly proposed methodology allows to perform inference of both structural and dynamical properties of diffusion on graphs, with better performance than state-of-the-art methods.

Strengths: - The paper proposes clear mathematical proofs for the construction of a new methodology to diffusion processes on graphs - Authors propose a new algorithm for studying diffusion processes on graphs, and demonstrate, on both synthetic and real-world data, that their method achieve better performance than state-of-the-art methods.

Weaknesses: The authors named there method Neural Mean-Field Dynamics but here the notion of neural network is defined in a very generic sense (discrete non-linear dynamical system of order 1, cf equation (14) ) far away from the common conception of RNN used in ML. Moreover it seems that the intervention of neural networks in the whole procedure is just one out of many insights on which the method relies. As such I wonder whether this will be of interest to the NeurIPS community and I think this work, of extraordinary quality, would be more suited for publications in network science conferences for instance.

Correctness: Claims, method and methodology look correct.

Clarity: The paper is well written.

Relation to Prior Work: Yes

Reproducibility: Yes

Additional Feedback: Update: I thank the authors for their response. However I still see a large distance between the common notion of RNN and the one used here, as such I still have my concern about the relevance for the NeurIPS audience and keep my score at 5. ---------------------- - In a typical RNN, inputs can be time varying signals, here what is termed inputs are simply the initial conditions of the dynamical system. - Can equations 14 be mapped on standard RNN architectures such as vanilla RNN, or LSTM, and interpret the x and h in terms of hidden state, gates or other ? - Could the statement lines 170-171 be expanded: with a more precise definition of \epsilon rather than « remaining nonlinear part « and take more words to define the « first layer of the RNN« ?


Review 2

Summary and Contributions: This paper proposes to model the diffusion on a network with a mean-field framework, which is derived from the Mori-Zwanzig formalism. The hard-to-solve memory term in the Mori-Zwanzig equation is approximated by a differential neural network. The differential equation is solved with Euler-forward discretization.

Strengths: i) The problem considered in this work is of high relevance. ii) Application of the Mori-Zwanzig formalism is in general appealing and novel. The resulting differential Equation (Thm. 2) is of interesting type and offers many future research directions. iii) The experimental section underlines the claims of interpretable results (Table 1).

Weaknesses: i) Many design choices appear rather arbitrary in section 3.2, e.g. why the split between linear and nonlinear parts for \epsilon(\cdot)? Since \epsilon is a black box, can we not get rid of the linear part from the beginning? ii) The connection to optimal control (Sec. 3.2) is confusing and seems out of context. a) Learning of weights as an optimal control problem is not new and was e.g. discussed in “Maximum Principle Based Algorithms for Deep Learning”. Such an optimal control connection can be done for any parameter inference problem. b) Why is Pontryagin maximum principle (PMP) introduced? From my understanding, the parameters are trained by minimizing loss function 18a with gradient descent. However if PMP is used, the solution quality heavily depends on the chosen solver method to the PMP problem, e.g. shooting methods. A discussion about such is missing. If PMP is not used (especially line 206 suggests no use of PMP), I would rather propose to get rid of this chapter, since it does not strengthen the main claims of the paper: Deriving a Mean Field framework from Mori-Zwanzig formalism. iii) How come in Fig. 1 the MAE of NMF decreases with increasing time horizon? I would expect the error to increase with increasing horizon, as it does for InfluLearner. The proposed model in this paper is an autoregressive one (Eq. 14a, b) and (small) errors at the beginning should accumulate to larger errors at later time steps. iv) Minor: Broader Scope just repeats the Intro/Conclusion. --- POST REBUTTAL --- I read the author response and the other reviews carefully. The main two issues I've raised, namely i) the paper is hard to follow ii) weak motivation of the central Eq. 10 in the paper, are shared by the other reviewers. I agree with the author’s comment that one could find some clues throughout the paper why Eq. 10 is in this particular shape, but this is rather hard to do (due to hard to read paper). Furthermore, I still wonder how the paper benefits from relation between standard-backprop and the Hamiltonian. I believe that the evaluation is correct, but why should this be in the paper? I would recommend the authors to rewrite the paper and motivate clearer the central design choices. In addition, I agree with Reviewer #2 that the paper would benefit more from being published at a network science conference. Consequently, I keep my score unchanged.

Correctness: The derivations and theoretical claims seem correct.

Clarity: i) The paper is rather hard to read and many terms/symbols are used without proper introduction, e.g. in Thm 1. z(t)=[x(t); e(t)] used without introducing e(t). ii) Experimental section is missing further details, e.g. initialization? Early stopping?

Relation to Prior Work: • The related work section is properly written. • The novelty of this work is clear.

Reproducibility: No

Additional Feedback: The theoretical contribution of this paper is valuable. However the writing of the paper of this paper does not meet the quality standards. I suggest rewriting and resubmitting the paper to an upcoming venue.


Review 3

Summary and Contributions: In this paper, authors propose the neural mean-field dynamics (NMF) framework to solve the prediction (future infection states of nodes) and network inference (connectivity and strength of impact between nodes) problem. Specifically, they use the Mori-Zwanzig formalism to derive a generalized Langevin equation (GLE). Further, this GLE is approximated by a deep neural network. Experimental results show that the proposed model outperforms existing approaches in different tasks.

Strengths: 1. The investigated problem is important and interesting. In addition, the proposed method is technically sound with theoretical proofs. 2. Evaluation is implemented on multiple generated datasets with different graph models and different distributions, and a real dataset from Sina Weibo. 3. This proposed model does not require early adopters or network structures, and the network structures can be inferenced based on the proposed model, which is different from the most existing methods.

Weaknesses: 1. The utilization of the approximation in (10) is not properly validated. For example, the error between the approximation of the deep neural network and the original Mori-Zwanzig memory term is not evaluated. 2.In the section of numerical experiments, different baselines are compared in different tasks. However, choosing them in these tasks is not well justified. For example, InfluLearner is only compared in the task of Infection probability and influence function estimation. Obviously, by combining with the classical greedy algorithm, it can be compared in the task of Influence Maximization. Thus, why choosing these compared algorithms in different tasks needs more discussion. 3. Technical details in this paper is a bit hard to follow. It is better to given a neural network diagram or a pseudo-code algorithm to help readers between understand the details of the proposed framework. 4. In line 216, it is said that 1,000 source sets are generated. However, in line 224, MAE is only averaged over 100 source sets, which is contrary to previous description. 5. There are many typos in this paper, e.g., - Line 40: “and and reture” - Line 127 and Line 142: “Appendix ??”

Correctness: Yes, both claims and method are correct.

Clarity: Most of them are well written, however, technical details are difficult of fellow.

Relation to Prior Work: yes

Reproducibility: Yes

Additional Feedback:


Review 4

Summary and Contributions: This paper studies diffusion in various networks. The main problem studied is the inference and estimation of when a particular node is infected and with what probability.

Strengths: The paper is generally well written and I understood the main motivations and contributions quite well. I am not in the best position to judge the competitiveness or the novelty of the method, however.

Weaknesses: A general comment from an outside perspective- Can you include in the introduction how your results would affect networks encountered in daily life like social networks or disease infection networks? In what ways does knowledge of timing and probability of infection affect me in my daily usage of social networks? In other words, describe some real-world applications of your results.

Correctness: I have not checked this.

Clarity: Yes.

Relation to Prior Work: I cannot judge this.

Reproducibility: No

Additional Feedback: I am an emergency reviewer for this paper, which, unfortunately is not at all in my field. So I am not in the best position to rate this paper. I have provided some general comments from an outsider's perspective. My rating as is, is based on the fact that the paper felt "pleasing" to read, and I understood the main motivations quite well. Since I cannot judge the novelty or the accuracy of the method, I have assumed these factors to be rated highly. ============================================================== POST-REBUTTAL: After the author response and other reviews, I am keeping my score unchanged.

[Author Response · NeurIPS 2020]

**Reviewer #2**

– Generic NN formulation and relation to RNN: Our (14) is a highly structured neural network (NN) due to the Mori-Zwanzig moment closure (please note the form of $\boldsymbol{f}$ in Eq.(7)). This enables us to learn the diffusion parameter and estimate influence (both for the diffusion network) using the mean-field dynamical system (14). This is different from learning a generic RNN which is not interpretable and does not serve the purpose of learning diffusion parameters.

– Relevance to NeurIPS: Learning diffusion parameters and predicting influence are one of central problems in the ML community. Please see [14-17],[21-23].

– Input is only the initial condition: This is because we need to estimate the influence of source set during testing phase, where no additional temporal data is available. Hence we can only use the source set as input of our network during training and testing. This is different from the standard RNN applications.

– Map to standard RNN/LSTM: Yes, we can map a dynamical system to other types of RNNs. However, our purpose is to derive a structured dynamical system for influence prediction, and need to integrate the diffusion parameters in the dynamical system for training in this work.

– A more precise definition of $\varepsilon$: in the present work, we set it to a generic NN to approximate the MZ memory kernel. We will state this with more details if a revision is allowed.

**Reviewer #4**

– Splitting the linear and nonlinear part in $\varepsilon$: This is because the linear part is a consequence of the kernel approximation of $h$ in Eq.(10) and $\boldsymbol{K}$ in Theorem 3. In this case, we can interpret the matrices $\boldsymbol{B}$ and $\boldsymbol{C}$ (and impose proper regularization or prior information if possible). The remainder part of $\varepsilon$ is set to generic NN, but not interpretable anymore.

– Connection to PMP: Unlike the standard optimal control and Ref.[37], our formulation does not allow time-varying control variables. This requires a modification of the PMP to interpret our network training. By introducing the total Hamiltonian, we showed that the standard back-propagation is directly equivalent to maximizing the total Hamiltonian. This is different from Ref.[37] which relies on successive approximation (different from back-propagation) and is computationally more expensive,

– Error of NMF decrease as time increase: this is because as time goes, all nodes will eventually be infected (if the diffusion network is connected). Therefore, all methods (not only NMF) will have lower prediction error as time goes to infinity. But NMF appears to be more accurate in predicting the error in early to middle stage of the propagation (rather than asymptotical error), which is of most interests in practice (e.g., one would like to predict the spread of news in three days rather than a month).

– $e(t)$ is not defined: $e(t)$ is defined in Eq.(5) above Theorem 1.

**Reviewer #5**

– Evaluation of approximation error: please note that it is *infeasible* in practice to evaluate the original MZ memory term, because exact evaluation involves solving an ODE system of size $2^n$ ($n$ is the size of the diffusion network). Instead, we use MC simulations to approximate ground truth as reference, and compared our results against it, as shown in our experiment part. This has been the standard approach for comparison in the literature.

– Using InfluLearner in influence maximization: Our comparison with InfluLearner on influence prediction demonstrated the significant improvement in accuracy. Therefore its performance cannot exceed NMF in influence maximization which heavily relies on the accuracy of influence prediction. Moreover, the computation complexity of InfluLearner is much higher than NMF, and not suitable for large scale problems and dense networks in influence maximization.

– Pseudo-code: the NMF network structure is given in Eq.(14), which can be solved by standard network training. We will provide more details if a revision is allowed.

– MAE only on 100 source sets: the 100 source sets refers to the testing data, whereas 1,000 refers to the training data.

**Reviewer #6**

– Examples: the network platform (like Twitter), campaign company, or advertiser are often interested in the influence of individuals (users) on networks—if the individual posts a news/advertisement, then it will be seen and retweeted by his/her followers, and then the followers of them, and so on. Accurate influence estimation proposed in our work helps the campaign company or advertiser to decide which individuals to select, at a certain monetary cost, to spread the news as fast as possible (to maximal amount of people on the network). It also has applications in disease intervention where

[Meta-Review · NeurIPS 2020]

The paper investigates how, in a diffusion process, the term accounting for the influence of the whole past, can be modelled in terms of temporal convolution and approximated via a recurrent neural network. The AC thinks that this topic is fully relevant to NeurIPS, and would be of utmost interest for an (admittedly small) fraction of the audience. However, the authors must make every effort to make their work accessible (even for statistical physicists; the Mori-Zwanzig fomalism is perhaps not as well known as the authors think, to say the least) and to thoroughly show how scalable the approach is compared to alternatives. The AC dearly hopes that the authors will invest in the pedagogical and writing efforts required to make their work known to the ML community. =========== Additional emergency review In this paper the authors predict an epidemic process on an unknown network by combining the Generalized Langevin equation (GLE) based on the Mori-Zwanzig formalism, together with deep learning. The context is typically the propagation of information (eg a tweet) on a social network, and the method assume fully observed cascade data, but not necessarily the network structure. This problem is difficult, especially if the network is not known. The originality of the approach lies in the combination of the GLE formalism which involves a memory kernel, with recurrent networks. The idea being to learn a stochastic equation which involves a vector of single node infection probabilities and where the effect of all the complex nodes interactions induced by the unknown network are encoded into the memory kernel thanks to the Mori-Zwanzig formalism. A crude approximation allows to learn the memory kernel in the form of a recurrent network from the data making the model easier to learn than a brute force LSTM based on cascade data. The latent state of the recurrent network is here simply obtained by applying a delay kernel to the previous configurations. Also the parameter learning of this model (i.e. the memory kernel and the underlying network parameters) is presented with an original point of view thanks to a reformulation in terms of an optimal control problem. I find some merit in establishing (i) a bridge between the seemingly far apart domains of stochastic differential equations and deep learning, (ii) a connection between parameter learning and optimal control which suggests some connection of the procedure with reinforcement learning; I found elegant the way the complexity of the process is summarized mainly in a delay kernel K(t-s, w) in (10) which then determines the latent state of the recurrent network. I found the numerical tests convincing, as they significantly and consistently improve on state of the art methods on various datasets to predict infected nodes but also to recover the network structure Albeit difficult to read because of the diversity of the material involved, I found this paper very interesting and stimulating. The approach seems quite general and possibly transferable to other types of processes, typically partially observed Markov processes. Remarks: I found a bit misleading the introduction of variables e_I in (5) since closed form and simpler equations can be obtained only in terms of variables x_I(t)